# Energy conservation by collective movement in schooling fish

**Yangfan Zhang\*, George V Lauder**

Department of Organismic and Evolutionary Biology, Harvard University, Cambridge, United States

**Abstract** Many animals moving through fluids exhibit highly coordinated group movement that is thought to reduce the cost of locomotion. However, direct energetic measurements demonstrating the energy-saving benefits of fluid-mediated collective movements remain elusive. By characterizing both aerobic and anaerobic metabolic energy contributions in schools of giant danio (*Devario aequipinnatus*), we discovered that fish schools have a concave upward shaped metabolism–speed curve, with a minimum metabolic cost at ~1 body length s$^{-1}$. We demonstrate that fish schools reduce total energy expenditure (TEE) per tail beat by up to 56% compared to solitary fish. When reaching their maximum sustained swimming speed, fish swimming in schools had a 44% higher maximum aerobic performance and used 65% less non-aerobic energy compared to solitary individuals, which lowered the TEE and total cost of transport by up to 53%, near the lowest recorded for any aquatic organism. Fish in schools also recovered from exercise 43% faster than solitary fish. The non-aerobic energetic savings that occur when fish in schools actively swim at high speed can considerably improve both peak and repeated performance which is likely to be beneficial for evading predators. These energetic savings may underlie the prevalence of coordinated group locomotion in fishes.

**\*For correspondence:**
yangfan_zhang@fas.harvard.edu

**Competing interest:** The authors declare that no competing interests exist.

## eLife assessment

The authors provide an **important** series of metabolic measurements characterizing group dynamics in fish, rationalizing that schooling behavior presents several benefits. The strength of evidence supporting this conclusion is **solid**, but the specific methodological and analytical approaches taken should be considered for further interpretation.

## Introduction

Newton's laws of motion underpin animal locomotion, from fine-scale maneuvers to long-distance migrations (*Biewener and Patek, 2018*). As speed increases or during migration involving sustained movement over long distances, animals often move in coordinated groups: *e.g.*, migratory birds in V-formation (*Weimerskirch et al., 2001*), ducklings swimming in formation (*Fish, 1995*), cyclists in a peloton (*Blocken et al., 2018*), elite marathon runners (*Ito, 2007*), and fish schools (*Weihs, 1973*). To overcome gravitational or fluid dynamic resistance to forward motion, animals use both aerobic and glycolytic metabolism (oxidative and substrate-level phosphorylation at the cellular level) to generate energy and sustain movement in the air, on land, and in the water. In water, a fluid that is 50 times more viscous than air and contains much less $O_2$ per Kg than air, the need for aquatic animals to reduce fluid dynamic drag for energy conservation is even greater than for aerial or terrestrial locomotion. Hence, we use fish schooling behavior as a model system to explore whether the fluid dynamics of collective movement (active and directional movement of animal groups along a common trajectory *Zhang and Lauder, 2023*) can enable energetic savings compared to locomotion by a solitary

**eLife digest** Schools of fish, flocks of birds flying in a V-formation and other collective movements of animals are common and mesmerizing behaviours. Moving as a group can have many benefits including helping the animals to find food and reproduce and protecting them from predators.

Collective movements may also help animals to save energy as they travel by altering the flow of air or water around individuals. Computational models based on the flow of water suggest several possible mechanisms for how fish swimming in schools may use less energy compared to fish swimming on their own. However, few studies have directly measured how much energy fish schools actually use while they swim compared to a solitary individual.

Zhang and Lauder used a device called a respirometer to directly measure the energy used by small tropical fish, known as giant danio, swimming in schools and on their own in an aquatic treadmill. The experiments found that the fish swimming in schools used 53% less energy compared with fish swimming on their own, and that fish in schools recovered from a period of high-speed swimming 43% quicker than solitary fish.

By adjusting the flow of the water in the tanks, the team were able to study the fish schools swimming at different speeds. This revealed that the fish used more energy when they hovered slowly, or swam fast, than when they swam at a more moderate speed. Previous studies have found that many fish tend to swim at a moderate speed of around one body length per second while they travel long distances. Zhang and Lauder found that the giant danio used the least energy when they swam at this 'migratory' speed.

These findings show that swimming in schools can help fish save energy compared with swimming alone. Along with furthering our understanding of how collective movement benefits fish and other animals, this work may help engineers to design robots that can team up with other robots to move more efficiently through the water.

individual. Fish schooling behaviors are some of the most prominent social and group activities exhibited by vertebrates.

Fish may school for many reasons, including foraging, reproduction, and migration, and fish schools often exhibit high-speed movement during evasion from predators. Hence, natural selection can put pressure on the formation of schools and individual interactions within a school. Besides 'safety in numbers' where individuals in a large school have a lower probability of becoming prey, can there be more tangible energetic benefits of high-speed swimming as a school? Since fluid drag scales as velocity squared (*Vogel, 1981*), movement at higher speeds places a premium on mechanisms to conserve energy. Fast and unsustainable swimming is fueled by aerobic metabolism with a major contribution from anaerobic glycolysis (*LaForgia et al., 2006*; *Brett, 1964*). During high-speed swimming aerobic capacity is maximized (*Brett, 1964*), and energy use is more likely to compete with other activities than in low-speed swimming which only occupies a small portion of aerobic capacity. The direct consequences of unsustainably engaging anaerobic glycolysis include both fatigue and metabolic perturbation (such as changing blood acidity) that need time to recover after bouts of intense locomotion. Animals become vulnerable during recovery because of their hindered ability to repeat peak locomotor performance in the presence of predators.

Another factor of importance in the experimental evaluation of the energetic cost of collective locomotion in fishes is that endurance and efficiency are often key to achieving lifetime fitness, especially in fish species that undergo significant migrations. Fish groups usually undergo long-distance migrations at slow speeds ranging from 0.25 to 1.0 BL s$^{-1}$ as recorded by tags on migrating fish (*Appendix 1—figure 1*). The common migratory speed of ~1 BL s$^{-1}$ for marine and anadromous fish could be related to the minimum aerobic cost of transport (*Beamish, 1978*). However, there is currently no direct measurement of the absolute cost of teleost fish locomotion demonstrating a minimum group metabolic rate ($\dot{M}O_2$) at an optimal speed ($U_{opt}$). Fish schools in nature swim over a wide range of speeds (*Appendix 1—figure 1*) but no study has directly characterized the total energy expenditure (TEE), and accounted for the potentially substantial anaerobic metabolic costs involved in higher speed locomotion. As a result, the swimming TEE performance curve of fish schools from the minimum swimming speed to the critical swimming speed ($U_{crit}$) has not yet been measured.

More broadly, despite the widespread notion that collective motion saves energy (*Weihs, 1973*), very few studies have directly measured the energetic cost of collective movement compared to the cost of movement by solitary individuals. In the canonical case of V-formation flight in birds, energetic saving is exclusively inferred from indirect measurements, for example, heart rate (*Weimerskirch et al., 2001*), and flapping frequency and phase (*Portugal et al., 2014*; *Table 1*). There are no direct measurements of metabolic energy consumption of collective movement in birds, and indeed this would be extremely challenging to accomplish. Although some energetic aspects of schooling fishes have been studied (*Parker, 1973*; *Abrahams and Colgan, 1985*; *Burgerhout et al., 2013*; *Currier et al., 2021*; *Hvas and Oppedal, 2019*), these studies have analyzed a limited range of speeds and focused on aerobic metabolism (see *Table 1*). In fact, several field and laboratory kinematic studies suggest that both bird flocks and fish schools do not always conserve aerobic energy (*Hvas and Oppedal, 2019*; *Partridge and Pitcher, 1979*; *Usherwood et al., 2011*) and indicate that moving in a group can actually involve *increased* costs. Without directly measuring the energetic cost of movement, inferences solely based on kinematics do not include complex group interactions, and the interaction between collective behavior and fluid dynamics. Consequently, we do not yet have a holistic and mechanistic view of where and how the collective movement by fish might conserve energy.

For swimming fish, fluid dynamic experiments have shown how collective movement can improve swimming efficiency due to interactions among deforming bodies and through interactions between moving animals and the local fluid environment (these energy-saving mechanisms are summarized in *Figure 1*). Our current understanding of energy saving mechanisms during collective locomotion in fishes is largely based on computational fluid dynamic models with a few analyses using robotic systems (*Weihs, 1973*; *Li et al., 2020*; *Kelly and Menzer, 2023*; *Fish et al., 2016*; *Kurt and Moored, 2018*) at low speed, but the link between such models and fish metabolic energy saving over a wide speed range is not yet established. We expect that the need to conserve energy in animals moving against air or water resistance should be greater at higher speeds. We hypothesize that fish in schools can reduce the total cost of high-speed locomotion relative to solitary movement, leading to reduced non-aerobic energetic costs and time of recovery for schooling fish swimming at speeds occupying the majority (>50%) of their aerobic capacity. Our secondary objective is to determine the differences (if any) between the swimming performance curves (metabolic rate versus speed) of solitary fish and fish schools.

To evaluate these hypotheses, we directly measured both the aerobic and non-aerobic energy used by schooling fishes over a wide range of water velocities (0.3–8.0 body lengths s$^{-1}$; BL s$^{-1}$), and then compared the swimming performance curve to that of solitary fish. We equipped a high-resolution (see materials and methods for our approach to enhancing the signal-to-noise ratio in our respirometer) swim-tunnel respirometer with two orthogonal high-speed cameras to quantify three-dimensional (3-D) fish kinematics. This enabled us to simultaneously measure energetics, 3-D dynamics of fish schools (n=5 replicate schools), and the kinematics of solitary fish (n=5) of a model species, giant danio (*Devario aequipinnatus*), that exhibits an active directional group swimming from near still water to maximum sustained speeds (equivalent to a Reynolds number range of 6.4•10$^3$–1.8•10$^5$; *Appendix 1—figure 2*), using a $U_{crit}$ test.

## Results

We discovered that both solitary fish and fish schools have a concave upward shaped aerobic metabolic rate ($\dot{M}O_2$)-speed curve over the lower portion of their speed range (0.3–3 BL s$^{-1}$, *Figure 2C, D*). The aerobic locomotor cost at 0.3–3 BL s$^{-1}$ showed no statistical difference between solitary fish and fish schools ($F_{1,80} \leq 2.40$, $p \geq 0.13$). Fish schools swimming at ~1.0 BL s$^{-1}$ consume less energy than at slower speeds ($F_{9,40} = 24.7$, p≤0.0007), while swimming at 3 BL s$^{-1}$ consumes a similar amount of energy to maintain position at a water velocity of 0.3 BL s$^{-1}$ (p=0.85; *Figure 2D*). Danio schools have a minimum aerobic cost ($\dot{M}O_{2min} = 212.9$ mg $O_2$ kg$^{-1}$ h$^{-1}$) of 1.25 BL s$^{-1}$. $\dot{M}O_{2min}$ at this optimal speed ($U_{opt}$) was *lower* than both the $\dot{M}O_2$ of aggregating behaviors (328.5 mg $O_2$ kg$^{-1}$ h$^{-1}$) exhibited before $U_{crit}$ test (*Appendix 1—figure 2*) and for the group at rest in still water $\dot{M}O_2$ (267.9 mg $O_2$ kg$^{-1}$ h$^{-1}$) (by 35% and 21% respectively; $F_{2,12} = 16.7$, p≤0.02, *Figure 2A, B*). Swimming at $U_{opt}$ reduces the energetic cost below that of swimming at either slower or higher speeds. Indeed, the Strouhal number of fish schools decreased from 2.1 at the lowest speed to 0.3 at the energetic minimum $U_{opt}$, and then increased to 0.4 at the higher $U_{crit}$.

**Table 1.** A summary of experimental studies that directly estimate the energetic saving of group movement in aerial and aquatic vertebrates that move through a fluid environment.

No energetic measurements have made for freely-moving bird flocks. Three studies measured the energetics of fish schools over a range of narrow speeds; two other studies measured the energetics of fish schooling at one speed. But no studies have quantified *both* the aerobic and anaerobic energetic cost of swimming as a group compared to solitary locomotion.

| Animal group | Species | Testing speeds | Group size | Measurements | Results | Reference |
|---|---|---|---|---|---|---|
| Bird | *Pelecanus onocrotalus* | Voluntary flights | <8 | Heart rate | Group movement saves energy | *Weimerskirch et al., 2001* |
| Bird | *Columba livia* | Voluntary flights | 6 | Flap frequency & Accelerations | Group movement costs more energy | *Usherwood et al., 2011* |
| Bird | *Geronticus eremita* | Voluntary flights | 14 | Wing beat phase, spatial distribution | V-formation birds use phasing strategy to cope with wakes | *Portugal et al., 2014* |
| Ray-finned fish | 15 species | 1 speed, unknown speed | 6–12 individuals | Aerobic energy | Grouping reduces metabolic rate | *Parker, 1973* |
| Ray-finned fish | *Notropis heterodon* | 1 speed ~1.2 BL s⁻¹ | 3 individuals | Aerobic energy | Only schooling of 6 cm fish saves energy | *Abrahams and Colgan, 1985** |
| Ray-finned fish | *Anguilla anguilla* L | 6 speeds 1–2.2 BL s⁻¹ | 7 individuals | Aerobic energy | Schooling save energy at low speeds | *Burgerhout et al., 2013** |
| Ray-finned fish | *Lepomis macrochirus Oncorhynchus mykiss* | 4 speeds, 1.5–3 BL s⁻¹ | 1–8 individuals | Aerobic energy | Schooling saves energy | *Currier et al., 2021** |
| Ray-finned fish | *Salmo salar* | 6 speeds 1–3 BL s⁻¹ | 10 individuals | Aerobic energy | Schooling does not save energy | *Hvas and Oppedal, 2019* |
| Ray-finned fish | *Devario aequipinnatus* | 14 speeds, 0.3–8 BL s⁻¹ | 8 individuals | Aerobic and anaerobic energy | Schooling saves energy at high speed | Present study* |

Other uncited studies of fish schoolings use kinematic calculations and computer simulation to indirectly show that fish schooling conserve energy.

*Studies used group and individual fish in the same size swim-tunnel respirometer.

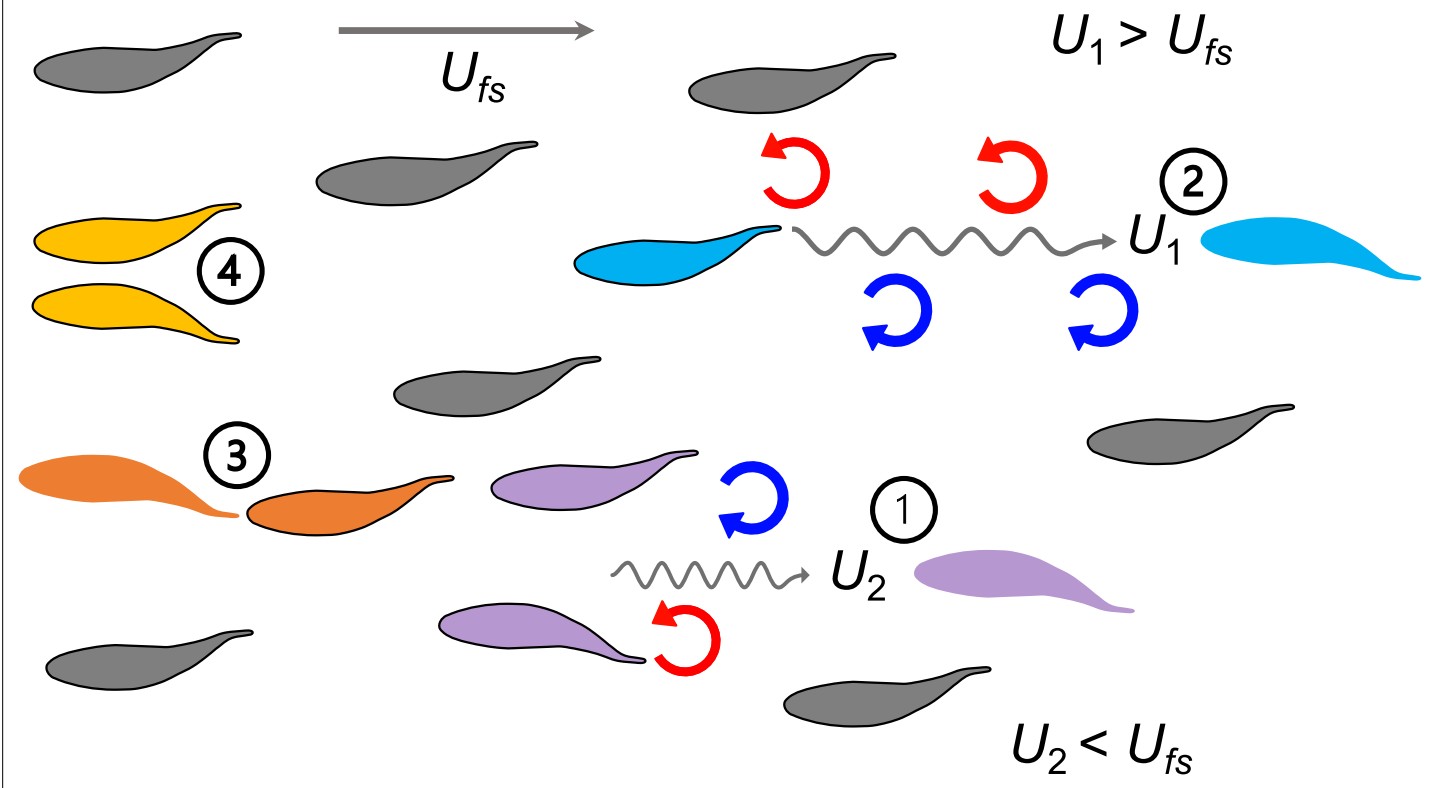

**Figure 1.** A summary of biomechanical principles underlying proposed hydrodynamic advantages of schooling behavior in fish. When fish swim into free-stream flow ($U_{fs}$), experimental data show that fish schools are dynamic with fish changing position frequently. Regardless of fish position within a school and changes in relative position, theoretical and robotic analyses have demonstrated that at least four mechanisms (indicated by numbers) provide an advantage in the form of reduced power consumption. 1. Reduced oncoming velocity ($U_2$) requires less thrust for a fish swimming in the wake between two leading fish (***Weihs, 1973***); 2. The Knoller-Betz effect of leading edge suction reduces costs for a trailing fish due to accelerated oscillating flow at the head ($U_1$) (***Jones et al., 1998***; ***Saadat et al., 2021***); 3. Added mass 'push' from follower to leader can reduce costs for the leader in front of another fish (***Fish and Hui, 1991***; ***Kurt and Moored, 2018***; ***Saadat et al., 2021***); 4. Wall effects benefit neighboring fish where swimming next to another fish reduces swimming costs (***Daghooghi and Borazjani, 2015***; ***Li et al., 2020***). These principles suggest that regardless of the relative positions of the individuals within the fish school, the fish school as a collective unit should be able to save metabolic energy compared to a solitary fish swimming in $U_{fs}$.

We also characterized the kinematic features that result in elevated $\dot{M}O_2$ at low water velocities. Danio maintaining body position in low-velocity aggregations had a higher tail beat frequency ($f_{TB}$) than solitary fish ($F_{1,80} \geq 9.8$, p≤0.002). However, at $U_{opt}$ and the highest speed of active directional schooling, fish in schools had a lower $f_{TB}$ ($F_{1,80} \geq 4.6$, p≤0.035) than solitary fish (***Figure 3B***). The 3-D angular heading of schooling fish transitioned from omnidirectional to pointing against water flow when water velocity is above 0.75 BL s$^{-1}$, and individual fish orient into the flow starting at 0.3 BL s$^{-1}$ (***Figure 3C***). Although the body angle and turning frequency of solitary fish and schools decreased with water velocity (***Video 1***), schooling fish had a higher turning frequency ($F_{1,80} \geq 15.6$, p<0.002) below 0.75 BL s$^{-1}$ than solitary fish (***Figure 3D, E***).

We discovered that, across the entire 0.3–8 BL s$^{-1}$ range, $\dot{M}O_2$-speed curves of fish schools are concave upward shaped (reached 1053.5 mg O$_2$ kg$^{-1}$ hr$^{-1}$ at 8 BL s$^{-1}$) whereas solitary fish showed an upward concave curve that reaching a plateau of 760.2 mg O$_2$ kg$^{-1}$ hr$^{-1}$ (~10% CV in 6–8 BL s$^{-1}$, ***Figure 2C***), demonstrating that schooling dynamics results in a 44% higher maximum aerobic performance ($F_{1,80} = 30.0$, p<0.001). This increased maximum aerobic performance translated to a lower use of aerobic capacity compared to solitary fish over 0.3–8 BL s$^{-1}$ (see the discussion for how fish swimming in school can improve aerobic performance). Collectively, fish schools used a 36% lower proportion of their aerobic scope than solitary fish (Wilcoxon test: $p = 0.0002$, ***Figure 2F***). Fish schools used a 38% lower proportion of aerobic scope (34 vs 55%) at 4 BL s$^{-1}$ (50% $U_{crit}$ and > 50% aerobic

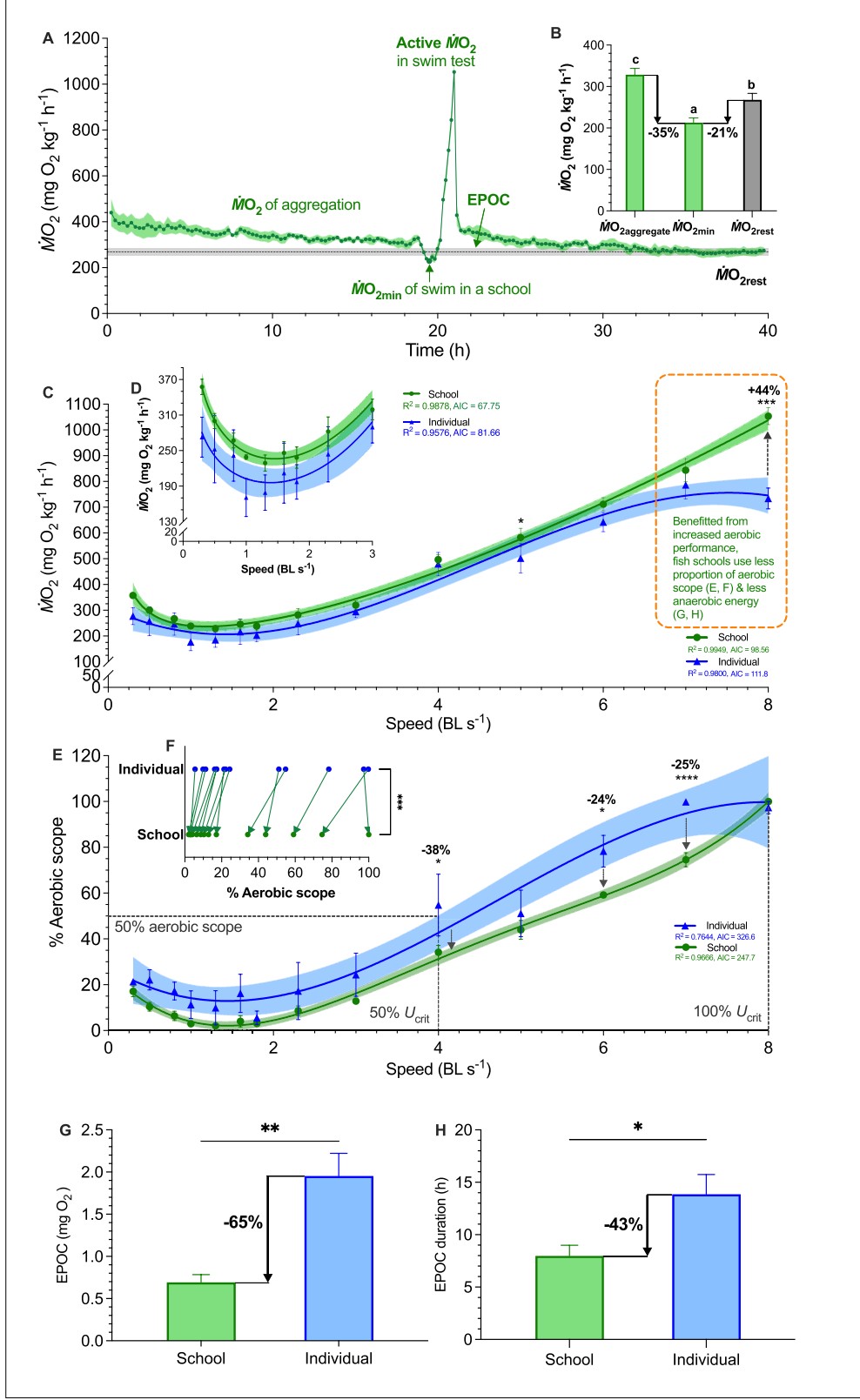

**Figure 2.** Measurements of aerobic and anaerobic locomotor cost of fish schools and solitary fish. (**A**) Average traces of metabolic rate ($\dot{M}O_2$) of fish schools over a 40 hr experiment. Following the first 18 hr quiescent state, a critical swimming speed ($U_{crit}$) test quantifies the aerobic cost of active swimming. The ensuing 18 hr measurement of excess post-exercise oxygen consumption (EPOC) quantifies the anaerobic cost. (**B**) Comparison of $\dot{M}O_2$ for

*Figure 2 continued on next page*

*Figure 2 continued*

conditions of aggregating behavior, minimum demand speed, and resting condition with minimal flow ($\dot{M}O_{2aggregate}$, $\dot{M}O_{2min}$, $\dot{M}O_{2rest}$) (**C**) Comparisons of concave upward shaped $\dot{M}O_2$-speed curve over the entire range (0.3–8 body length s$^{-1}$, BL s$^{-1}$) and (**D**) the concave shaped $\dot{M}O_2$-speed curve at the lower speeds (0.3–3 BL s$^{-1}$). (**E, F**) Percentage (%) aerobic scope used by fish schools and solitary fish during the $U_{crit}$ test. (**G, H**) Comparisons of EPOC and EPOC durations between fish schools and solitary fish. Statistical significance is denoted by asterisk(s). Green color = school data (n=5 schools); blue color = solitary fish data (n=5 individual fish); shading indicates the 95% confidence interval. Statistical details are available in the statistical analyses section.

scope onward), and consistently used ~25% lower proportion of their aerobic scope at 6 and 7 BL s$^{-1}$ than solitary fish ($F_{1,103} \geq 4.8$, p≤0.03, *Figure 2E*).

Given that fish schools had a higher maximum aerobic performance, generally used a lower proportion of their aerobic scope, and individuals within the schools have a 14% lower $f_{TB}$ than the solitary fish at 8 BL s$^{-1}$ (11.5 vs. 13.3 Hz, $F_{1,80} = 15.1$, p<0.001), we predicted that fish schools, compared with solitary fish, use less anaerobic energy to supplement aerobic energy for the high-speed movement approaching their aerobic limit. Hence, we measured post-exercise O$_2$ utilization, a majority of which is used to restore high-energy phosphate storage and glycolytically induced metabolic perturbations (EPOC: Excess Post-exercise O$_2$ Consumption). We discovered that fish schools have a 65% lower EPOC (0.69 vs 1.95 mg O$_2$; $t_8$=4.5, p=0.0021), and recover 43% faster than solitary fish (8 vs 14 h; $t_8$=2.8, p=0.025; *Figure 2G, H*).

To estimate the relative proportions of aerobic and anaerobic energy contributions for locomotion at each swimming speed, we modeled EPOC in addition to $\dot{M}O_2$, the aerobic cost (>50% $U_{crit}$ and aerobic scope, *Figure 4A*; *Supplementary file 1b*). We also estimated the total O$_2$ cost during the entire swimming process for each school and individual and calculated total energy expenditure (TEE) (*Brett, 1964*). The TEE of fish schools was 38–53% lower than that of the solitary fish between 5 and 8 BL s$^{-1}$ ($F_{1,103} \geq 7.4$, p≤0.008; *Figure 4D*). TEE of fish schools was only 42–143% higher than the aerobic metabolic rate at 5–8 BL s$^{-1}$ ($F_{1,96} \geq 3.5$, p≤0.001) and anaerobic metabolic energy only accounted for 29–58% of the TEE depending on speed (*Figure 4B*). In contrast, TEE of solitary fish was 131–465% higher than the aerobic metabolic rate between 5–8 BL s$^{-1}$ ($F_{1,112} \geq 10.7$, p≤0.001), where anaerobic energy accounted for 62–81% of the TEE depending on speed (*Figure 4C*).

Schooling dynamics reduced the total cost (aerobic plus anaerobic) of transport (TCOT) by an average of 43% compared to swimming alone ($F_{1,103} = 6.9$, p=0.01), and most of this energy conservation happens at higher speeds when fish approach their aerobic limit. Schooling dynamics in danio enables an extremely shallow rate of increase in TCOT with speed compared to that of an individual (*Figure 4E*). The TCOT of solitary fish increased by 490% (6.5 kJ km$^{-1}$ kg$^{-1}$) at 8 BL s$^{-1}$, whereas the schooling TCOT increased by only 200% (3.6 kJ km$^{-1}$ kg$^{-1}$) at 8 BL s$^{-1}$. Therefore, individual fish form a more energy-efficient biological entity when they collectively move as a school.

To answer the question of how schooling dynamics reduces TEE, we combine video analysis of fish tail beat kinematics with simultaneous aerobic and anaerobic measurements to compute energy expended *per tail beat* (TEE•beat$^{-1}$), and compared values for fish in schools to those for solitary fish. Schooling fish reduced TEE•beat$^{-1}$ by 30–56% at higher speeds compared to solitary fish ($F_{1,81} \geq 7.3$, p≤0.008), a substantial reduction in TEE•beat$^{-1}$ consumed both by the school, and by individual fish within a school (*Figure 3A*). Notably, the energetic benefits during active directional schooling occur when fish schools become more streamlined, as the length of fish school increased with speed and plateaued beyond 2 BL s$^{-1}$ (*Figure 3F*), while the 3-D distance among individuals stayed relatively constant at ~1.2 BL (*Figure 3G*). These results directly demonstrate schooling dynamics benefits the swimming kinematics of individual fish within the school and results in a net outcome of up to 53% TEE reduction in fish schools compared to solitary fish.

## Discussion

Simultaneous characterization of energetics and kinematics enables an integrated understanding of both the physiology and physics of fish schooling behavior. Hydrodynamic models, kinematic measurements, and robotic analyses of fish schools indicate that the cost of swimming can be reduced when fish swim beside neighboring fish (*Li et al., 2020*), behind or in front of another fish

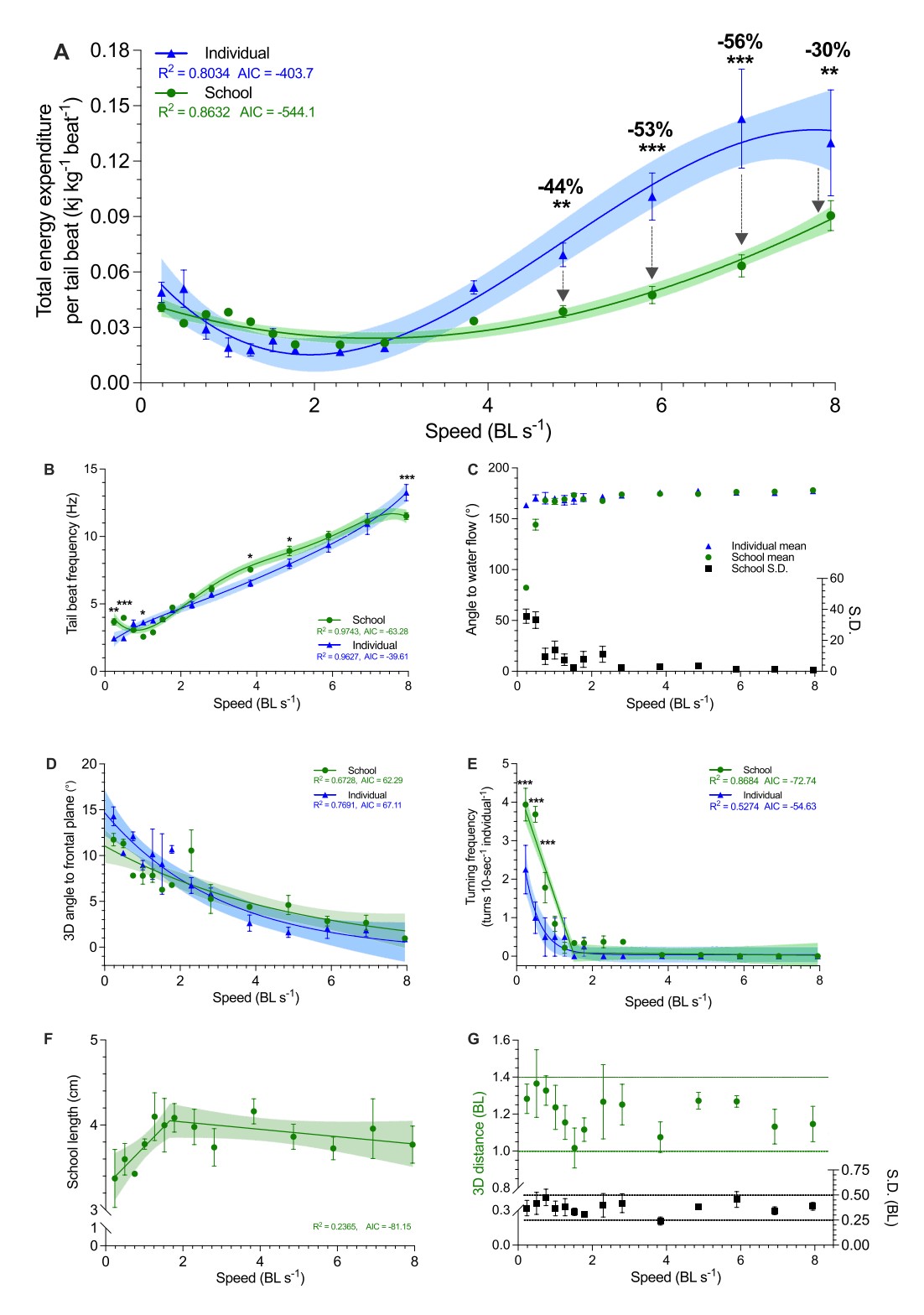

**Figure 3.** Three-dimensional characterization of swimming kinematics and fish schooling dynamics as a function of speed. (**A**) Total energy expenditure (TEE) per tail beat, (**B**) Tail beat frequency, (**C**) The angle of fish to free-stream water flow, measured as the mean and the S.D. of the angles of the individuals within the school. (**D**) Three-dimensional angle of fish to the frontal plane. (**E**) Turning frequency, (**F**) Three-dimensional school length. (**G**) Three-dimensional distances among all individuals in the school and the S.D. of the distance. The upper and lower boundaries of the metrics are indicated. Statistical significance is denoted by asterisk(s). Green color = school data (n=3–4); blue color = solitary fish data (n=3–4); shading indicates the 95% confidence interval. See materials and methods for details of three-dimensional reconstruction and statistics.

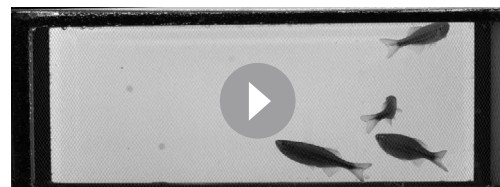

**Video 1.** A high-speed video of stability control of a small fish school at near-still water velocity. Three of the four fish that were holding stationary showed a positive body angle, and considerable motion of the fins is evident for stability control. One fish that moved from the top right corner increased its body angle and started to use more fin movements for stability control as the fish approached zero ground speed. Once the fish reached zero ground speed, the body angles and movements of fins were very similar to the other three fish that had been holding stationary.

https://elifesciences.org/articles/90352/figures#video1

(*Jones et al., 1998*), or behind leading individuals (*Weihs, 1973*; *Figure 1*). We have demonstrated here that fish schools downshifted an overall concave upward swimming energetics performance curve at higher speeds by ~43%. Energy use as swimming speed increases changes in a non-linear manner, with higher energy use at the lowest speeds (*Figure 2D*), and the performance curve below 3.0 BL s⁻¹ is *U*-shaped. Energy saving occurs through a substantial reduction in the non-aerobic energy contribution when fish approach aerobic limits. One of the key ecological benefits of the reduced use of glycolysis is a faster recovery time from fatigue (*Figure 2H*). This would enable fish schools in nature to repeat high-performance movements. Considering that high-speed maneuvers are extremely common in predatory evasion (*Stewart et al., 2018*), food-searching group motion (*Pitcher et al., 1982*), and in fish schools in the ocean (*Appendix 1—figure 1*), energy savings by fish schooling at high speed and the faster recovery that followed could have a considerable impact on lifetime fitness (*Jain and Farrell, 2003*; *Holder et al., 2022*). Moreover, considerable energy savings occur even though we observed that individual danio regularly change positions within the school and do not maintain stable inter-individual locations. We expect that this is because individuals within the school can use multiple hydrodynamic mechanisms to save energy depending on their location (*Figure 1*). We propose that energetic savings by collective movement may not require either fixed positional arrangements among individuals or specific kinematic features (such as tail beat synchronization) compared to solitary locomotion. Based on these results, we regard fish schooling as a highly robust behavior that provides considerable energetic benefits that are not necessarily sensitive to specific fish locations or movements. Future studies are necessary to decipher each of the specific hydrodynamic mechanisms for energy saving.

## Schooling dynamics enhances aerobic performance and reduces non-aerobic energy use

We discovered that a significant amount of energy conservation for active directional swimming occurs at speeds above 3 BL s⁻¹. In nature, fish schools routinely exhibit active directional collective locomotion above ~6 BL s⁻¹ (*Appendix 1—figure 1*; *Supplementary file 1c*; *Misund and Aglen, 1992*), a speed that engages anaerobic glycolysis. Yet, there are no previous measurements of the anaerobic cost of schooling in fish. Hence, important and previously unrecognized benefits of active directional collective locomotion are (1) increased aerobic performance, (2) a reduced use of aerobic capacity and anaerobic energy, and (3) a resultant faster recovery from the associated metabolic perturbations and costs of swimming at higher speeds. When animals are approaching their maximum metabolic rate, the highest attainable rate of O₂ uptake limits aerobic performance during the unsteady state of high-speed locomotion and anaerobic glycolysis is engaged to support the maximum metabolic demand.

Fish swimming at higher speeds (>50% $U_{crit}$) routinely exhibit a burst-&-glide gait mode of locomotion (*Peake and Farrell, 2006*; *Peake and Farrell, 2005*). Repeated bursting is substantially fueled by anaerobic glycolysis and the gliding phase enables peak $\dot{M}O_2$ (*Zhang et al., 2019*) to replenish the venous O₂ content (*Farrell and Clutterham, 2003*). Fish within a school are known to increase gliding time and decrease burst time by 19% when they are trailing the leading fish (*Fish et al., 1991*), *e.g*, a lower $f_{TB}$ of fish schools at 8 BL s⁻¹ is related to the extended gliding time (*Figure 4B*). We reason that the higher proportion of time spent gliding likely enables more bouts of peak $\dot{M}O_2$ and enhances the maximum aerobic performance (*Zhang et al., 2019*; *Farrell and Clutterham, 2003*). This enhanced maximum aerobic performance increases the metabolic ceiling and enables fish to use a lower proportion of their aerobic scope for locomotion (*Figure 2E*). Physiological studies of exercise metabolism

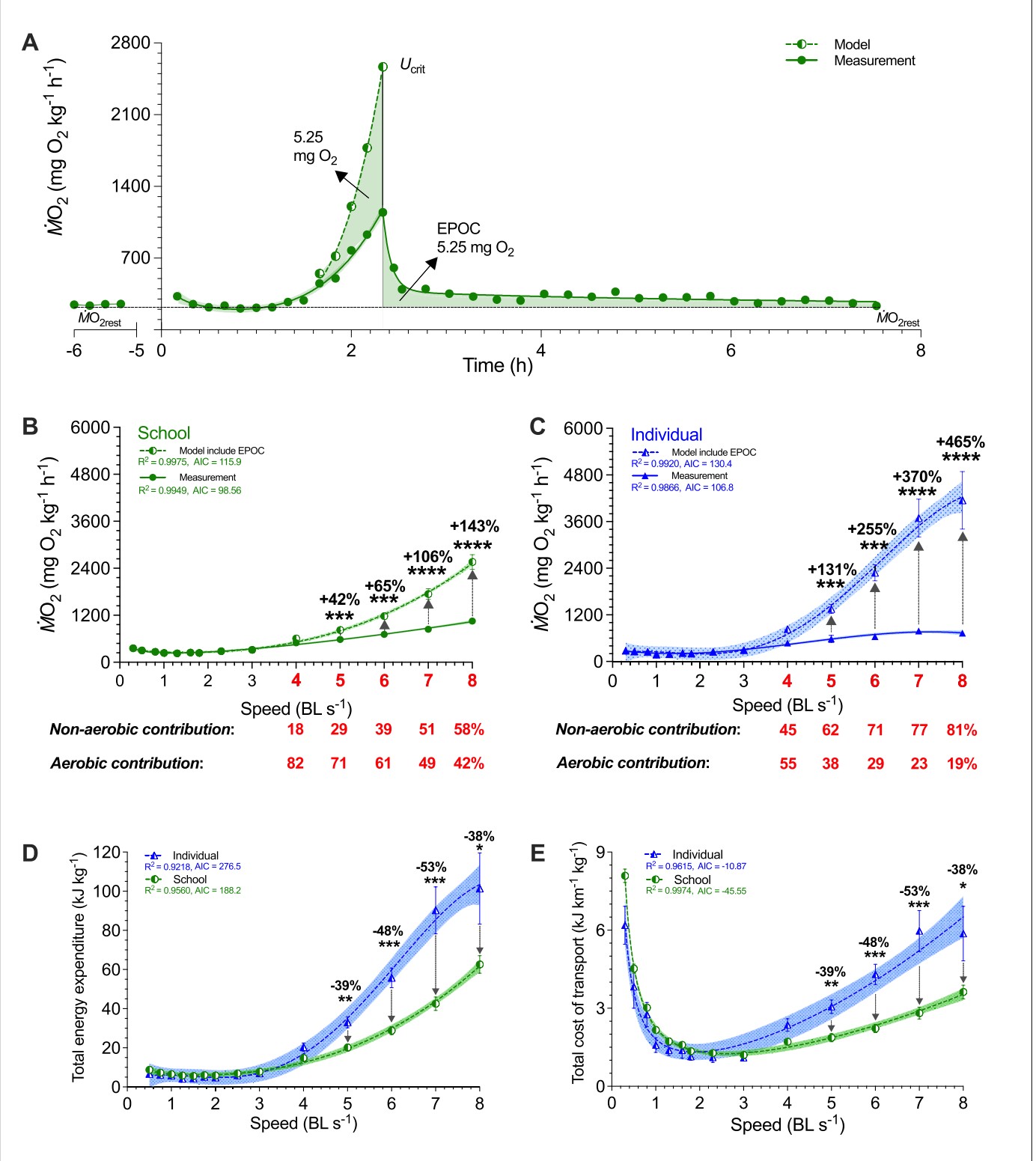

**Figure 4.** Modeling of simultaneous aerobic and non-aerobic costs of fish schools and solitary fish for a critical swimming speed ($U_{crit}$) test. (**A**) Modeling the $O_2$ cost of the metabolic rate ($\dot{M}O_2$)-speed curve and the ensuing recovery cost (excess post-exercise oxygen consumption, EPOC) as a function of speed. After $U_{crit}$, fish returned to the same resting $\dot{M}O_2$ ($\dot{M}O_{2rest}$) as a pre-test. (**B, C**) In addition to $\dot{M}O_2$ (solid line and filled symbols), we modeled the total $O_2$ cost (dash line and half-filled symbols) for fish schools and solitary fish and when performing the $U_{crit}$ swimming test. The estimated partitioning of aerobic and non-aerobic contributions to swimming are denoted (red-&-bold) with respect to speed for 4–8 BL s[-1] is shown below each graph.

*Figure 4 continued on next page*

*Figure 4 continued*

(**D, E**) Using total $O_2$ cost, we computed total energy expenditure (10 min period per point) and the total cost of transport (including both aerobic metabolism, high-energy phosphate, and anaerobic glycolysis) for both fish schools and solitary fish. Statistical significance is denoted by asterisk(s). Green color = school data (n=5); blue color = solitary fish data (n=5); shading indicates the 95% confidence interval. See materials and methods for modeling and statistical details.

and locomotion suggest that a lower proportional use of aerobic scope during movement relates to a reduced accumulation of anaerobic end-products (*LaForgia et al., 2006*). Collectively, our results showed that the increased aerobic performance (*Figure 2C*) and the reduced use of the aerobic capacity (*Figure 2E*) for swimming above 50% $U_{crit}$ in fish schools likely plays a key role in reducing the use of glycolysis and the accumulation of lactate (e.g. EPOC, *Figure 2G*) for high-speed swimming (*Brett, 1964*; *Peake and Farrell, 2005*).

We present direct evidence of substantial non-aerobic energy saving by demonstrating the 65% lower EPOC (*Figure 2G*) used by fish that swam in schools. Anaerobic glycolysis is crucial in permitting continued movement when aerobic limits are reached at high swimming speeds (*Gaesser and Brooks, 1984*). Often vertebrates at higher speeds use more fast twitch fibres that generate high-frequency contractile force in part through anaerobic glycolysis (*Jayne and Lauder, 1996*; *Rome et al., 1985*). The higher EPOC for single fish is unlikely to have been confounded by any possible stress effect of swimming as a solitary fish. Stressed solitary fish would have elevated aerobic metabolic rates in low water velocity, which we show is not the case for our experiments (see *Figure 2D*). Indeed, to mitigate the stress response, we acclimated fish to the solitary conditions and habituated fish to the respirometer for a quiescent day before the experiment (see more details in experimental animal and protocols). When vertebrates move at high speed, the typical 'stress' neurotransmitters (e.g. adrenaline, catecholamines, and cortisol) usually increase (*Milligan, 1996*; *Kjaer et al., 1988*), regardless of whether vertebrates move as an individual or in a group. These endocrinological responses underpin the physiological mechanisms that enable higher aerobic performance (e.g. rising heart rate) (*Brown et al., 1979*; *Ross et al., 1984*). Moreover, after the highest swimming speeds immediately followed by an active recovery process, fish likely prioritize the recovery of the essential functions (as the levels of the neurotransmitters decline fairly rapidly) for the potential of repeated locomotion performance (speedy recovery allows fish to restore their swimming performance) (*Peake and Farrell, 2005*; *Zhang et al., 2018*; *Lee et al., 2003*).

To present a complete energetic profile showing the total energy expenditure (TEE) of locomotion for fish schooling, we model the non-aerobic cost (EPOC) on top of the aerobic swimming performance curve. The model is rooted in the use of aerobic and glycolytic energy when fish swim above 50% $U_{crit}$ (*Peake and Farrell, 2006*) and a commonly observed inflection point of faster anaerobic end-product accumulation is at ~50% aerobic scope (*LaForgia et al., 2006*; see materials and methods for detailed physiological bases and criteria for EPOC modeling). As a result, we observed a TEE being ~2.5-fold higher than the aerobic metabolic energy expenditure which agrees with the theoretical estimates based on individual yearling sockeye salmon (*Webb, 1975*). This model elucidates how aerobic and anaerobic metabolic energy constitute TEE, fuel muscles, generate locomotor thrust and overcome hydrodynamic drag during swimming.

We also demonstrated that the need for energy saving in fish schools at lower speeds (<3 BL s⁻¹ or 38% $U_{crit}$ *Figures 2E, 3D, E*) is not as crucial as at higher speeds where we demonstrated substantial energy savings in the schools compared to solitary individuals. Fluid drag is exponentially less at lower speeds compared to higher speeds. Fish predominately use aerobic metabolism to support low-speed steady swimming for prolonged periods (*Brett, 1964*) which does not require lengthy recovery as swimming at higher speeds, where glycolysis contributes to the energetic demands. The costs of low-speed swimming are often less than <20% of aerobic capacity (*Figure 2E*), which leaves the majority of aerobic capacity (>50%) for other activities. Therefore, the benefits of reducing the energetic cost of locomotion are likely not major factors underlying behaviors such as low-speed milling (*Costanzo and Hemelrijk, 2018*) and aggregation (*Rountree, 1989*) in fishes, where other ecological drivers such as feeding, predator avoidance, and reproduction likely dominate the fitness landscape.

## Schooling dynamics and energy conservation

Our 3D kinematic analyses shed light on the complex interactions between schooling and hydrodynamics that enable energy saving by fish collective movement. One of the key possible mechanisms is local interactions among individuals, where the kinematics of individual fish respond to the wakes shed by neighbors (*Camazine et al., 2020*). Kinematic and simulation studies indicate that two to three coordinated fish can save energy (*Verma et al., 2018*; *Li et al., 2020*) through local interactions. We directly demonstrate here that eight coordinated fish can save energy, and recent simulations show that schooling benefits can extend to at least 23 fish (*Kelly and Menzer, 2023*). Although these studies appear to suggest that the energetic benefits of schooling are scalable, we are still some ways from proving the exact mechanisms of how the large school size in nature is coordinated and whether or not local interactions are still one of the mechanisms for energy saving when scaling up school size. Future hydrodynamic and long-term 3-D tracking studies are needed to investigate the specific schooling formations and hydrodynamic mechanisms used by live fish for energy saving (as summarized in *Figure 1*; *Ko et al., 2023*).

We show that danio in a school keep a relatively consistent 3-D distance from neighbors and have small variation in mean position even as individuals routinely change position within the school (*Figure 3G*). Computational fluid dynamic analyses of fish schools show that a small 3D neighboring distance (e.g. <0.4 BL s$^{-1}$) increases drag and can increase swimming costs, whereas a large 3-D neighboring distance (e.g. >2 BL s$^{-1}$) reduces the hydrodynamic benefits (*Verma et al., 2018*). There may thus be an optimal mean distance among individuals within a school to maximize energetic savings. Whether inter-individual distance should change as swimming speed increases remains an open question and may be a function of school size. While we cannot completely preclude the possibility that elongation of danio fish school might be due to the weaker individuals falling behind, the fact that the school maintained a stable 3-D distance as speed increased and that fish continued to change position within the school suggests that the elongated school volume at higher speeds may reflect changing hydrodynamic interactions among individuals (*Figure 1*).

To encapsulate the complex interaction among animal physiology, kinematics and fluid dynamics, more comprehensive quantification beyond simple kinematic metrics are necessary. For example, the wing flapping frequency of birds in a V-formation can be higher than for solitary flight despite strong indirect evidence of overall energy saving by V-formation flying (*Weimerskirch et al., 2001*; *Portugal et al., 2014*; *Usherwood et al., 2011*), and flapping phase of adjacent birds may be informative for understanding the fluid dynamic advantages of V-formation flying (*Portugal et al., 2014*). Likewise, phase matching of body motion between neighboring fish can help individual fish within the school to boost thrust or reduce drag (*Li et al., 2020*). Fish can also adjust their body stiffness and maintain $f_{TB}$ while reducing the amount of muscle activity needed to generate movement (*Beal et al., 2006*; *Liao et al., 2003*; *Quinn and Lauder, 2021*). Thus, we did not observe consistent and substantial changes in the $f_{TB}$ of fish within the schools compared to solitary swimmers despite demonstrating substantial energetic savings by the group at high speed. Although reduced $f_{TB}$ has been used in past studies as an indirect indicator of energy saving in fishes (*Zuyev and Belyayev, 1970*; *Marras et al., 2015*), we caution that $f_{TB}$ is not necessarily an indicator of energy use in group dynamics where fish constantly and dynamically interact with complex vortices and flow generated by other individuals. By simultaneously measuring kinematics and energetics, we discovered a downshifted performance curve of TEE per tail beat in fish schools at higher speeds (*Figure 3A*), even with limited alteration in $f_{TB}$ at lower swimming speeds within the school. This suggests that the locomotor muscle fibres in the body musculature of fish within the school need to generate less force as indicated by the lower measured TEE. Fish can possibly fine-tune undulatory motion to harness kinetic energy from nearby vortices (*Li et al., 2020*; *Verma et al., 2018*), reducing biological energy contribution for thrust.

## Aerobic metabolic rate–speed curve of fish schools

We discovered that the aerobic metabolic rate ($\dot{M}O_2$)–speed curve at speeds less than 3 BL s$^{-1}$ is $U$-shaped, with swimming at 0.3 BL s$^{-1}$ utilizing the same aerobic energy as moving at 3 BL s$^{-1}$ (*Figure 2D*). The entire aerobic energy-speed performance curve is concave upward in an extended $J$-shape (*Figure 2C*), and non-linear, in contrast to several previous analyses of energy use with speed in fish which have shown linear relationships (*Herskin and Steffensen, 1998*; *Bushnell et al., 1984*; *Sepulveda and Dickson, 2000*). By quantifying energy use over a wide speed range and by measuring

aerobic energy use at 14 distinct speeds, we showed that locomotion at the lowest speeds involves increased energy use relative to swimming at speeds up to 3 BL s$^{-1}$, similar to previous results from skate locomotion (*Di Santo et al., 2017*). This suggests that analyses of fish energy use as speed increases would benefit from increasing the number of tested speeds, and that efforts to extend speed measurements to both the lowest and highest speeds would enable a greater level of precision in measuring fish locomotor performance.

One key finding in this regard is that fish schools show a minimal absolute energetic cost ($\dot{M}O_2$) at a mean group swimming speed of ~1.0 BL s$^{-1}$ that is higher than the minimum swimming speeds tested. Thus, fish schools can swim at a speed with the least amount of energy use and potentially extend the distance travelled with the same amount of metabolic substrate onboard. This swimming speed of minimum cost closely matches the migratory speeds of carangiform and subcarangiform migratory fishes (often as schools) which are in the range of 0.5–1.5 BL s$^{-1}$ as recorded by tags on migrating fish (*Block et al., 1992*; *Block et al., 2005*; *Appendix 1—figure 1*; *Supplementary file 1a and c*). Although *D. aequipinnatus* is not a migratory species, our results suggest that the migratory speed of fish schools likely occurs at the speed showing the minimum aerobic metabolic rate for long-distance locomotion. Indeed, the Strouhal number when fish schools swim at $U_{opt}$, showing a minimum aerobic energy cost of locomotion, is 0.3, a hallmark of efficient swimming for many fish species (*Triantafyllou and Triantafyllou, 1995*). Fish of different size swimming at 1 BL s$^{-1}$ will necessarily move at different Reynolds numbers, and hence the scaling of body size to swimming speed needs to be considered in future analyses of other species that differ in size.

The exact energetic mechanisms underpinning the chosen migratory speeds of fish schools would benefit from more in-depth studies. Since both fish schools and solitary individuals have $U_{opt}$ of ~1 BL s$^{-1}$, the $\dot{M}O_2$–speed curve of a fish school might be the average of individual curves from fish within the school. However, pushing and pulling forces from hydrodynamic interactions of neighboring fish could help swimmers settle into stable arrangements, a phenomenon known as the Lighthill conjecture (*Kurt et al., 2019*). Further studies will need to explore whether individuals within the school are more frequently located at points recommended by the Lighthill conjecture which could potentially result in minimum metabolic costs of locomotion at $U_{opt}$ in fish schools (*Ko et al., 2023*). Research on fish swimming in large aquaculture circular tanks has discovered that holding a large school of salmonids at a water velocity of ~1 BL s$^{-1}$ resulted in healthy growth and good conditions (*Timmerhaus et al., 2021*). This phenomenon seems to suggest that the minimum metabolic costs of locomotion yield the largest available aerobic capacity for growth and other functions.

Higher locomotor costs at the lowest speeds are likely caused by higher postural costs for active stability adjustments in the near-still fluid. The direct link between higher energetic costs caused by a higher 3-D body angle and higher turning frequency at the lowest speeds in solitary danio (*Figure 4D, E*) supports previous results in skates (*Di Santo et al., 2017*) where energetic costs are high at the very low swimming speeds due to the increased energetic cost involved in maintaining body stability and generating lift in negatively buoyant fishes. We expect more species that move in the fluid to show a higher postural cost in near-still fluid but careful measurements of both body kinematics and position along with high-resolution respirometry and accurate low-speed flow control will be required to demonstrate this phenomenon in a diversity of fish species. Elevated costs should be indicated at the lowest speeds as a direct increase in $\dot{M}O_2$ and not just as COT. When the denominator (i.e. speed) for deriving COT is less than 1, COT tends to skew towards the higher value at the lowest speeds. Mechanistically, fish at low fluid speeds have lower muscle and propeller efficiencies (*Webb, 1975*) with reduced fluid-assisted stability. Increased active stability adjustments such as fin movements (*Di Santo et al., 2017*) and tilting behaviors that enable the fish's body to act as a hydrofoil (*He and Wardle, 1986*) are used for stability control (see *Video 1*). Moreover, danio do not show intermittent swimming gaits at low speed as observed in some labriform swimmers (*Cathcart et al., 2017*; *Gellman et al., 2019*), and hence gait-specific kinematics most likely do not play a role in the elevated locomotion cost that we observe at the lowest speeds. Also, the higher turning frequency at <0.75 BL s$^{-1}$ in fish schools can relate to the need for coordination with neighbors. Nevertheless, as fish move more rapidly over a speed range, we note that the total cost of transport of ~1.2 kJ km$^{-1}$ kg$^{-1}$ at 3 BL s$^{-1}$ is among the lowest recorded for aquatic organisms, and is less than the TCOT (1.5 kJ km$^{-1}$ kg$^{-1}$) for jellyfish (the most energetically efficient low-speed swimmers; *Blocken et al., 2018*) swimming at <2 BL s$^{-1}$ (*Xu and Dabiri, 2020*).

In summary, our experiments on giant danio have demonstrated substantial energy conservation resulting from schooling dynamics across a wide range of speeds in fish. Direct measurement of both aerobic and non-aerobic energy use is critical for understanding the rapid collective movement of animals. Fish schooling in the high-drag viscous aquatic medium serves as a model for understanding how group movement by animals can be a more energy-efficient biological collective than movement by isolated individuals. By increasing maximum aerobic performance, fish schools save anaerobic energy and reduce the recovery time after peak swimming performance. Furthermore, by decreasing the proportion of metabolic capacity and recovery time devoted to locomotion, animals can apportion more energy to other fitness-related activities, such as digestion, growth and reproduction. More broadly, comprehending how the collective dynamics of animal movements in the water, land, and air can modify the energy use profiles of individuals provide a better understanding of the ecological and evolutionary implications of group locomotion.

## Materials and methods
### Experimental animals
The experiments were performed on giant danio (*Devario aequipinnatus*) that were acquired from a local commercial supplier near Boston, Massachusetts USA (*Supplementary file 1*). Five schooling groups are randomly distributed and housed separately in five 37.9 l aquaria (n=8 per tank). The five solitary individuals are housed separately in five 9.5 l aquaria (n=1 per tank). The individual housing condition acclimated the single *D. aequipinnatus* to the solitary environment and helped to reduce any isolation stress that might elevate whole-organism metabolic rate. In fact, the aerobic locomotion cost of solitary individuals showed no statistical difference from (in fact, being numerically lower) that of fish schools at a very low testing speed. The flow speed is similar to some areas of the aerated home aquarium for each individual fish. This suggests that the stress of solitary fish likely does not meaningfully contribute to the higher locomotor costs (see experimental protocol for more details on mitigating the stress). The condition factor showed no difference between solitary fish and fish schools (0.81 vs 0.99; $t_8$=2.14, p=0.065). All aquaria have self-contained thermal control (28 °C), an aeration system (>95% air saturation, % sat.) and a filtration system. Water changes (up to 50% exchange ratio) were carried out weekly. Fish were fed ad libitum daily (TetraMin, Germany). Animal holding and experimental procedures were approved by the Harvard University Animal Care IACUC Committee (protocol number 20-03-3).

### Integrated Biomechanics & Bioenergetic Assessment System (IBAS)
The core of our Integrated Biomechanics & Bioenergetic Assessment System (IBAS) (*Appendix 1—figure 4*) is a 9.35 l (respirometry volume plus tubing) customized Loligo swim-tunnel respirometer (Tjele, Denmark). The respirometer has an electric motor, and a sealed shaft attached to a propeller located inside the respirometer. Regulating the revolutions per minute (RPM) of the motor controls water velocity of the tunnel. The linear regression equation between RPM and water velocity (V) is established (V=0.06169 • RPM – 5.128, $R^2$=0.9988, p<0.0001) by velocity field measured by particle image velocimetry (PIV). Hence, the aerobic costs during locomotion are measured through the regulation of the water velocity.

The swim-tunnel respirometer is oval-shaped. The central hollow space of the respirometer increases the turning radius of the water current. As a result, the water velocity passing the cross-section of the swimming section (80×80 × 225 mm) is more homogenous (validated by direct particle image velocimetry (PIV) of flow in the working section of the respirometer following the procedures in *Lee et al., 2022*). Moreover, a honeycomb flow straightener (80×80 × 145 mm) is installed in the upstream section of the swimming section that was specifically designed to create relatively uniform flow across the working section (also confirmed by direct flow imaging). Based on our flow visualization analyses, the effective distance of slow-moving fluid due to boundary layer was <2.5 mm at speeds above 2 BL s$^{-1}$. The boundary layer played an ever-diminishing role at higher speeds (>4 BL s$^{-1}$) when energy saving of fish schools becomes more predominant. Danio (~10 mm wide) cannot effectively hide in the narrow boundary layer created by our flow baffle system. In addition, the convex hull volume of the fish school did not change as speed increased, suggesting that the fish school was not flattening against the wall of the swim tunnel, a typical feature when fish schools are benefiting from

wall effects. In nature, fish in the centre of the school effectively swim against a 'wall' of surrounding fish where they can benefit from hydrodynamic interactions with neighbors.

To standardize the experimental apparatus and avoid instrument variation, solitary fish and fish schools are measured in the same swim-tunnel respirometer, and the same overall experimental design is used as in previous studies (*Abrahams and Colgan, 1985*; *Parker, 1973*; *Burgerhout et al., 2013*). The ratios of respirometer:individual volume ($r_{RI}$) in our experiments was 2200 for individual fish (we used larger solitary *D. aequipinnatus* to increase the signal-to-noise ratio), and 693 for fish schools. The $r_{RI}$ is essentially a balance between giving enough space for fish to exhibit natural loco-motion and reducing wall effects and generating a reliable signal-to-noise ratio to measure the decline of dissolved $O_2$ (DO) in water (*Svendsen et al., 2016*). The increase in the signal-to-noise ratio can come from the better technology of the $O_2$ probe (we used an optical $O_2$ probe which has a higher measurement sensitivity than the Winkler method or Electrode & Galvanic probes used in the previous studies, see *Zhang, 2021* for review) and modifications to the swim-tunnel respirometer. We used a water loop installed 95 cm downstream of the propeller with water returned to the respirometer 240 cm upstream of the swimming section. Flow in the water loop moves (produced by an in-line circulation pump, Universal 600, EHEIM GmbH & Co KG, Deizisau, Germany) in the same direction as the water flow in the swimming tunnel. A high-resolution fibre optic $O_2$ probe (Robust oxygen probe OXROB2, PyroScience GmbH, Aachen, Germany) is sealed in the loop in the downstream of the circu-lation pump where there is better mixing to continuously measure the DO level in the water (recording frequency ~1 Hz, response time <15 s). The designated water sampling loop together with the water mixing by the propeller and water pumps effectively reduces the noise-to-signal ratio. As a result, the respirometry system reaches a stable signal-to-noise ratio once the sampling window is longer than 1.67 min (see *Appendix 1—figure 5*), well within the duration of the velocity step to obtain a stable signal-to-noise ratio for calculating $\dot{M}O_2$ (*Zhang et al., 2019*).

The oxygen probe was calibrated to anoxic (0% sat., a solution created by super-saturated sodium sulphite and bubbling nitrogen gas) and fully aerated water (100% sat.). After fish removal, the back-ground $\dot{M}O_2$ in the swim-tunnel respirometer was measured for a 20 min sealed period before and after each trial to calculate the average background $\dot{M}O_2$ (<6% of fish $\dot{M}O_2$), which was used to correct for the $\dot{M}O_2$ of fish (*Svendsen et al., 2016*). The pre-filtered water (laboratory grade filtration system) is constantly disinfected by UV light (JUP-01, SunSun, China) located in an external water reservoir to suppress the growth of microbial elements. Water changes of 60% total volume occurred every other day and a complete disinfection by sodium hypochlorite is conducted weekly (Performance bleach, Clorox & 1000 ppm).

To simultaneously measure schooling dynamics and swimming kinematics, the customized oval-shaped swim-tunnel respirometer is located on a platform with an open window beneath the swim-ming section. The platform is elevated 243 mm above the base to allow a front surface mirror to be installed at a 45° angle. This mirror allows a high-speed camera (FASTCAM Mini AX50 type 170 K-M-16GB, Photron Inc, United States, lens: Nikon 50mm F1.2, Japan) to record the ventral view. The second camera (FASTCAM Mini AX50 type 170 K-M-16GB, Photron Inc, United States, lens: Nikon 50mm F1.2, Japan) is positioned 515 mm to the side of the swimming section to record a lateral view. Synchronized lateral and ventral video recordings were made at 125 fps, and each frame was 1024 by 1024 pixels. To avoid light refraction passing through the water and distorting the video record-ings, the swim-tunnel respirometry is not submerged in a water bath. Temperature regulation of the respirometer is achieved by regulating room temperature, installing thermal insulation layers on the respirometer and replenishing the water inside the respirometer from a thermally regulated (28 °C, heater: ETH 300, Hydor, United States & chiller: AL-160, Baoshishan, China) water reservoir (insulated 37.9 l aquarium) located externally.

The aerated (100% sat., air pump: whisper AP 300, Tetra, China) reservoir water is flushed (pump: Universal 2400, EHEIM GmbH & Co KG, Deizisau, Germany) to the respirometer through an in-line computer-controlled motorized ball valve (U.S. Solid) installed at the in-flow tube. The other in-line one-way valve is installed at the out-flow tube. The out-flow tube is also equipped with a one-way valve. The valve is shut during the measurement period, a precautionary practice to eliminate the exchange of water between the respirometer and the external reservoir when the water moves at a high velocity inside the respirometer. This flushing was manually controlled to maintain DO above 80% sat. Every time the respirometer was closed to measure $\dot{M}O_2$, the water temperature fluctuates no

more than 0.2 °C. The water temperature inside the respirometer is measured by a needle temperature probe (Shielded dipping probe, PyroScience GmbH, Aachen, Germany) sealed through a tight rubber port of the respirometer.

To allow fish to reach the undisturbed quiescent state during the trial (another stress mitigation practice), the entire IBAS is covered by laser blackout sheets (Nylon Fabric with Polyurethane Coating; Thorlabs Inc, New Jersey, United States). The room lights are shut off and foot traffic around IBAS is restrained to the absolute minimum. Fish are orientated by dual small anterior spots of white light (lowest light intensity, Model 1177, Cambridge Instruments Inc, New York, United States) for orientation (one to the top and the other to the side) of the swimming section. The test section is illuminated by infrared light arrays to allow high-speed video recording with minimal disturbance of fish behavior.

## Experimental protocol

The energy use of vertebrates at lower speeds is primarily aerobic, while for high-speed movement anaerobic metabolic pathways are activated to supply the additional (largely shorter-term) energy needs (**LaForgia et al., 2006**). While whole-animal aerobic metabolism is measured by oxygen ($O_2$) uptake rate ($\dot{M}O_2$), the non-aerobic $O_2$ cost (mostly through high-energy phosphate and anaerobic glycolysis) is measured as excess post-exercise $O_2$ consumption (EPOC; **Brett, 1964**). Both metabolic energy sources contribute to the total energy expenditure (TEE) required for movement. For a given workload, the higher the maximum aerobic performance of the animal, the less the need for the anaerobic energy contribution (**LaForgia et al., 2006**; **Gaesser and Brooks, 1984**). The experimental protocol captures both metabolic energy contributions by measuring $\dot{M}O_2$ during locomotion and EPOC afterwards.

We studied five replicate schools and five replicate individuals. Swimming performance test trials were conducted with *Devario aequipinnatus* fasted for 24 hr, a sufficient period for a small size species at 28 °C (i.e. high resting $\dot{M}O_2$) to reach an absorptive state. In fact, we observed no specific dynamic action, the amount of $O_2$ consumed for digestion during the first diurnal cycle (**Appendix 1—figure 3**). Prior to the swimming performance test, testing fish were gently weighed and placed in the swim-tunnel respirometer. The fish swam at 35% $U_{crit}$ for 30 min to help oxidize the inevitable but minor lactate accumulation during the prior handling and help fish become accustomed to the flow conditions in the swim-tunnel respirometer (**Lee et al., 2003**). After this time, the fish to be tested were habituated (>20 hr) to the respirometer environment under quiescent and undisturbed conditions. The long habituation period also helps to reduce the stress and further reduce the likelihood that the solitary fish might be more stressed than the fish schools and showed an elevated $\dot{M}O_2$ (**Parker, 1973**). During this time, we used an automatic system to measure the resting $\dot{M}O_2$ for at least 19 hr. Relays (Cleware GmbH, Schleswig, Germany) and software (AquaResp v.3, Denmark) were used to control the intermittent flushing of the respirometer with fresh water throughout the trial to ensure $O_2$ saturation of the respirometer water. $\dot{M}O_2$ was calculated from the continuously recorded DO level (at 1 Hz) inside the respirometer chamber. The intermittent flow of water into the respirometer occurred over 930 s cycles with 30 s where water was flushed into the respirometer and 900 s where the pumps were off and the respirometer was a closed system. The first 240 s after each time the flushing pump was turned off were not used to measure $\dot{M}O_2$ to allow $O_2$ levels inside the respirometer to stabilize. The remaining 660 s when the pumps were off during the cycle were used to measure $\dot{M}O_2$. The in-line circulation pump for water in the $O_2$ measurement loop stayed on throughout the trial.

We characterize the aerobic costs for the swimming performance of fish using an established incremental step-wise critical swimming speed ($U_{crit}$) test (**Brett, 1964**). The first preliminary trial determined the $U_{crit}$ of this population of *Devario aequipinnatus* as 8 BL s$^{-1}$. Characterizing the swimming performance curve required a second preliminary trial to strategically select 9 water velocities (0.3, 0.5, 0.8, 1.0, 1.3, 1.5, 1.8, 2.3, 2.8 BL s$^{-1}$) to bracket the hypothesized upward concave shaped aerobic metabolism-speed curve at the lower speed (<40% $U_{crit}$). Additional five water velocities (3.8, 4.9, 5.9, 6.9, 8.0 BL s$^{-1}$) are used to characterize the exponentially increasing curve to the maximum and sustained swimming speed, $U_{crit}$ (see **Appendix 1—figure 6**). Altogether, 14 points provide a reliable resolution to characterize the swimming performance curve. The cross-section of the danio school is ~10% of the cross-sectional area of the swim tunnel, hence the blocking effect is negligible (**Kline et al., 2015**). At each water velocity, fish swam for 10 min (**Di Santo et al., 2017**) to reach a steady state in $\dot{M}O_2$ at low speed (see **Appendix 1—figure 7**). Above 40% $U_{crit}$, $\dot{M}O_2$ can become more

variable (*Zhang et al., 2019*). Hence, in this protocol, we focus on measuring the sustained aerobic energy expenditure by calculating the average $\dot{M}O_2$ for each 10 min velocity step using *Equation 1*. At the 5th min of each velocity step, both ventral and lateral-view cameras are triggered simultaneously to record 10 s footage at 125 frames per second, at 1/1000 shutter speed and 1024×1,024 pixel resolution. Thus, both data streams of $\dot{M}O_2$ and high-speed videos are recorded simultaneously. The $U_{crit}$ test is terminated when 12.5% of fish in the school or a solitary individual touches the back grid of the swimming section for more than 20 secs (*Lee et al., 2003*). The $U_{crit}$ test lasted ~140 min and estimates the aerobic portion of TEE over the entire range of swimming performance.

To measure the contribution of non-aerobic $O_2$ cost, where the majority of the cost is related to substrate-level phosphorylation, and to calculate the TEE for swimming over the entire speed range, we measured EPOC after the $U_{crit}$ test for the ensuing 19 hr, recorded by an automatic system. Most previous measurements of EPOC have used a duration of ~5 hr (see review *Zhang et al., 2018*), but our extended measurement period ensured that longer duration recovery $O_2$ consumption during EPOC was measured completely as fish were exercised to $U_{crit}$ (see summary table in *Currier et al., 2021*). The intermittent flow of water into the respirometer occurred over 30 s to replenish the DO level to ~95% sat. For the following 900 s the flushing pump remained closed, and the respirometer becomes a closed system, with the first 240 s to allow $O_2$ saturation inside the respirometer to stabilize. The remaining 660 s when the flushing pump was off during the cycle were used to measure $\dot{M}O_2$ (see *Equation 1*). The cycle is automated by computer software (AquaResp v.3) and provided 74 measurements of $\dot{M}O_2$ to compute EPOC. Upon the completion of the 3-day protocol, the school or individual fish are returned to the home aquarium for recovery. The fish condition was closely monitored during the first 48 hr after the experiment, during which no mortality was observed.

## Bioenergetic measurement and modeling

To estimate the steady-rate whole-animal aerobic metabolic rate, $\dot{M}O_2$ values were calculated from the sequential interval regression algorithm (*Equation 1*) using the DO points continuously sampled (~1 Hz) from the respirometer.

$$\dot{M}O_2 = \left[ \frac{d_{DO\,[i,(i+a)]}}{d_{t\,[i,(i+a)]}} \bullet (V_r - V_f) \bullet S_O \right] / (t \bullet M_f) \tag{1}$$

where $\frac{d_{DO}}{d_t}$ is the change in $O_2$ saturation with time, $V_r$ is the respirometer volume, $V_f$ is the fish volume (1 g body mass = 1 ml water), $S_o$ is the water solubility of $O_2$ (calculated by AquaResp v.3 software) at the experimental temperature, salinity and atmospheric pressure, $t$ is a time constant of 3600 s h$^{-1}$, $M_f$ is fish mass, and $a$ is the sampling window duration, $i$ is the next $PO_2$ sample after the preceding sampling window.

To account for allometric scaling, the $\dot{M}O_2$ values of solitary fish were transformed to match the size of the individual fish in the school using an allometric scaling exponent (b=0.7546). The calculation of the scaling relationship [$Log_{10}(\dot{M}O_2)$=b•Log10(M)+Log10(a), where $M$ is the body mass and $a$ is a constant] was performed by least squares linear regression analysis (y=0.7546 • x+0.2046; $R^2$=0.6727, p<0.0001) on the 180 data points of metabolic rate and body mass from a closely related species (the best available dataset to our knowledge) (*Wootton et al., 2022*). (The mass scaling for tail beat frequency was not conducted because of the lack of data for *D. aequipinnatus* and its related species. Using the scaling exponent of distant species for mass scaling of tail beat frequency will introduce errors of unknown magnitude.) The allometrically scaled $\dot{M}O_2$ values were used to derive other energetic metrics (listed below and such as aerobic scope) for the solitary fish. The energetic metrics of fish schools are calculated from the mass-specific $\dot{M}O_2$.

The resting oxygen uptake ($\dot{M}O_{2rest}$), the minimum resting metabolic demands of a group of fish or a solitary individual, is calculated from a quantile 20% algorithm (*Chabot et al., 2016*) using the $\dot{M}O_2$ estimated between the 10–18th hr and beyond the 32nd –51st hr of the trial. These are the periods of quiescent state when fish completed the EPOC from handling and swimming test.

The $\dot{M}O_2$ for the aggregation (*Figure 2B*, $\dot{M}O_{2aggregate}$) is calculated as the average value using the $\dot{M}O_2$ estimated between the 10 and 18th hr without the effects of any tests.

Minimum oxygen uptake ($\dot{M}O_{2min}$) is the lowest $\dot{M}O_2$ value recorded in the entire trial, which always occurred at the optimal speed when a school of fish collectively reached the lowest $\dot{M}O_2$ value.

Active oxygen uptake ($\dot{M}O_{2active}$) is the highest average $\dot{M}O_2$ when fish are actively swimming (**Brett, 1964**).

The aerobic scope (the metric for aerobic capacity) is the numerical difference between $\dot{M}O_{2active}$ and $\dot{M}O_{2min}$ (i.e. $\dot{M}O_{2active} - \dot{M}O_{2min}$) (**Brett, 1964**).

The percentage of aerobic scope (% aerobic scope) is calculated by normalizing the $\dot{M}O_2$ value at a water velocity as a % aerobic scope: $[(\dot{M}O_2 - \dot{M}O_{2min}) / (\dot{M}O_{2active} - \dot{M}O_{2min})] * 100\%$. The apportioning of aerobic scope to swimming performance is computed across the entire range of swim speeds.

The excess post-exercise oxygen consumption (EPOC) is an integral area of $\dot{M}O_2$ measured during post-exercise recovery, from the end of $U_{crit}$ until reached $\dot{M}O_{2rest}$ plus 10% or within 24 hr post-exercise, whichever endpoint occurred first. This approach reduces the likelihood of overestimating EPOC due to any potential spontaneous activities. To account for the allometric scaling effect, we used the total amount of $O_2$ consumed (mg $O_2$) by the standardized body mass of fish (1.66 g) for fish schools and solitary fish.

We model EPOC (i.e. non-aerobic $O_2$ cost) and $\dot{M}O_2$ during $U_{crit}$ to estimate a *total* $O_2$ cost over the duration of the swimming performance test. Our conceptual approach was pioneered by **Brett, 1964** in fish (**Svendsen et al., 2010**) and is also used in sports science (**LaForgia et al., 2006**). Mathematical modeling of EPOC and $\dot{M}O_2$ during swimming was applied to study the effects of temperature on the total cost of swimming for migratory salmon (**Lee et al., 2003**). We improved the mathematical modeling by applying the following physiological and physics criteria. The first criterion is that significant accumulation of glycolytic end-product occurred when fish swimming above 50% $U_{crit}$ (**Peake and Farrell, 2005**) which corresponds to >40% $\dot{M}O_{2max}$ (or ~50% aerobic scope) (**LaForgia et al., 2006**). This is also when fish start unsteady-state burst-&-glide swimming gait (**Peake and Farrell, 2005**). The second criterion is that the integral area for the non-aerobic $O_2$ cost during swimming can only differ by ≤0.09% when compared to EPOC. The non-aerobic $O_2$ cost during swimming is the area bounded by modeled $\dot{M}O_2$ and measured $\dot{M}O_2$ as a function of time when fish swim >50% $U_{crit}$ (see **Figure 2A**). The third criterion is that total energy expenditure is expected to increase exponentially with swimming speed (**Appendix 1—figure 8**). Specifically, these curves were fitted by power series or polynomial models, the same models that describe the relationship between water velocity and total power and energy cost of transport (**Appendix 1—figure 8**). Following these criteria, the non-aerobic $O_2$ cost at each swimming speed is computed by a percentage (%) modifier based on the aerobic $O_2$ cost (**Supplementary file 1b**). The exponential curve of total $O_2$ cost as swimming speed of each fish school or solitary individual was derived by an iterative process until the difference between non-aerobic $O_2$ cost and EPOC met the 2nd criterion. The sum of non-aerobic $O_2$ cost and aerobic cost gives the total $O_2$ cost.

Total energy expenditure (TEE) is calculated by converting total $O_2$ cost to kJ ×kg$^{-1}$ using an oxy-calorific equivalent of 3.25 cal per 1 mg $O_2$ (**Brafield and Solomon, 1972**).

Cost of transport (COT), in kJ ×km$^{-1}$×kg$^{-1}$ is calculated by dividing TEE by speed (in km ×hr$^{-1}$; **Di Santo et al., 2017**).

## Three-dimensional kinematic data extraction from high-speed videography

We used two synchronized 10 s high-speed videos (lateral and ventral views, at each speed) for kinematic analyses. We calibrated the field of view of the high-speed cameras using a direct linear transformation for three-dimensional (3-D) kinematic reconstruction (DLTdv8; **Hedrick, 2008**) by applying a stereo calibration to the swimming section of the respirometer (see **Appendix 1—figure 9**). We digitized the anatomical landmarks of fish (see **Appendix 1—figure 10**) to obtain the X, Y, and Z coordinates for each marker at the 1st s, 5th s, and 10th s for videos recorded at each speed. These coordinates are used to calculate the following kinematic parameters. All the calculations are validated on the known length and angle of test objects inserted into the tank working section.

The 3-D distance between the tip of the nose of each fish in the school per frame is calculated in vector **Equation 2**

$$d = \sqrt{(Xa - Xb)^2 + (Ya - Yb)^2 + (Za - Zb)^2} \qquad (2)$$

where spatial coordinates of two neighboring fish are ($X_a$, $Y_a$, $Z_a$) and ($X_b$, $Y_b$, $Z_b$) respectively.

The 3-D angle of each fish in the school to the frontal plane per frame is calculated by *Equation 3*

$$\theta = \arccos \frac{(X2 - X1) \cdot (X3 - X1) + (Y2 - Y1) \cdot (Y3 - Y1) + (Z2 - Z1) \cdot (Z3 - Z1)}{\sqrt{(X2 - X1)^2 \cdot (Y2 - Y1)^2 \cdot (Z2 - Z1)^2} \cdot \sqrt{(X3 - X1)^2 \cdot (Y3 - Y1)^2 \cdot (Z3 - Z1)^2}}$$ (3)

where the spatial coordinates of the caudal peduncle, nose of fish and right-angle crosshair between the peduncle and the nose are $(X_1, Y_1, Z_1)$, $(X_3, Y_3, Z_3)$ and $(X_2, Y_2, Z_2)$ respectively (see *Appendix 1—figure 9*).

The fish's angle to water flow per frame is calculated using the arctangent function in Excel (Microsoft, United States) using the spatial coordinates of the caudal peduncle and nose of the fish.

The school length (in *X*-axis) is calculated as a 3-D distance (*Equation 2*) between the nose of the first fish of the school to the caudal peduncle of the last fish in the school.

The values (3-D distance of the individuals, 3-D body angle, 3-D school length) calculated above were averaged among the three frames at each speed as a representative kinematic feature of the schools (or solitary fish) for each speed. The standard deviation of the angle to water flow in fish schools is calculated from the angular values of the individuals.

Tail beat frequency ($f_{TB}$), in Hz, is calculated as the number of tail beats observed within 1 s. We sampled 10 tail beats from fish exhibiting steady-state swimming to obtain an average and representative $f_{TB}$ for each solitary fish at each speed. We sampled 10 tail beats of steady-state swimming for three individuals in the center of the fish school for an average and representative $f_{TB}$ for each fish school.

Turning frequency is the total number of events when fish made a 90° turn in the 10 s video. To make the total number of turns per individual and fish schools comparable (multiple individuals inevitably have a larger absolute number of turns), we derived the average turning frequency per schooling fish by dividing the total number of turns of fish school by the number of fish in the school.

Total energy expenditure per tail beat (kJ kg$^{-1}$ beat$^{-1}$) is calculated by TEE $\bullet$ ($f_{TB} \bullet 60$)$^{-1}$ to estimate the total metabolic energy spent to achieve one stride length.

Additional calculations of fluid dynamic metrics were:

Strouhal number = ($f_{TB} \bullet$ tail beat amplitude) $\bullet$ $U^{-1}$ (*Vogel, 1981*). Tail beat amplitude is measured as the average distance of five peak-to-peak oscillation amplitudes of the tip of the fish's tail. The measurement is conducted on the calibrated high-speed video in video analysis software (Phontron FASTCAM Viewer 4, Photron USA, Inc).

Reynolds number = (water density $\bullet$ $U$ $\bullet$ fish fork length) $\bullet$ water dynamic viscosity $^{-1}$ (*Vogel, 1981*). Water density and dynamic viscosity are given at 28 °C.

## Statistical analyses

Measurement points are presented as mean ± s.e.m. For the metrics that failed normality tests, logarithm transformations were applied to meet the assumptions of normality of residuals, homoscedasticity of the residuals, and no trend in the explanatory variables. Since fish schools exhibit the features of a coherent functional unit, the statistical model treated one school as one sample size, and the experiment measured five replicates of fish schools. The majority of statistical comparisons used the General Linear Model (Fixed factors: solitary fish vs. fish schools and swimming speeds, the label of solitary fish and fish schools as a random effect) with Holm–Šídák post-hoc tests. The few comparisons that used other statistical models are listed below. The comparison of $\dot{M}O_{2min}$ at $U_{opt}$, $\dot{M}O_2$ of aggregating behaviors exhibited at the lowest speed ($\dot{M}O_{2aggregate}$) and resting $\dot{M}O_2$ ($\dot{M}O_{2rest}$) in fish schools used one-way ANOVA with Holm–Šídák post-hoc tests. The comparison of EPOC between solitary fish and fish schools used a two-tailed Student's t-test. The comparison of the duration of EPOC between solitary fish and fish schools used a two-tailed Student's t-test. The overall difference in percentage aerobic scope (% aerobic scope) between fish schools and solitary individuals is compared by the Wilcoxon singed-rank test over the entire range of 14 swimming speeds. The statistical analyses were conducted in SPSS v.28 (SPSS Inc Chicago, IL, USA). The best-fitting regression analyses were conducted using Prism v.9.4.1 (GraphPad Software, San Diego, CA, USA). 95% C.I. values were presented for all regression models as shaded areas around the regression or data points. Statistical significance is denoted by *, **, ***, **** for p-values of ≤0.05, ≤0.01, ≤0.001, ≤0.0001 respectively.

## Acknowledgements

Many thanks to members of Lauder Laboratory for numerous discussions about fish schooling behaviour, for comments on the manuscript, and to Cory Hahn for fish care. Funding provided by the National Science Foundation grant 1830881 (GVL), the Office of Naval Research grants N00014-21-1-2661 (GVL), N00014-16-1-2515 (GVL), N00014-22-1-2616 (GVL), and a Postdoctoral Fellowship of Natural Sciences and Engineering Research Council of Canada PDF-557785-2021 (YZ).

## Additional information

### Funding

| Funder | Grant reference number | Author |
| --- | --- | --- |
| National Science Foundation | 1830881 | George V Lauder |
| Office of Naval Research | N00014-21-1-2661 | George V Lauder |
| Office of Naval Research | N00014-16-1-2515 | George V Lauder |
| Office of Naval Research | N00014-22-1-2616 | George V Lauder |
| Natural Sciences and Engineering Research Council of Canada | PDF-557785-2021 | Yangfan Zhang |

The funders had no role in study design, data collection and interpretation, or the decision to submit the work for publication.

### Author contributions

Yangfan Zhang, Conceptualization, Resources, Data curation, Software, Formal analysis, Funding acquisition, Validation, Investigation, Visualization, Methodology, Writing - original draft, Writing – review and editing; George V Lauder, Resources, Supervision, Funding acquisition, Project administration, Writing – review and editing

### Author ORCIDs

Yangfan Zhang ⓘ http://orcid.org/0000-0001-5625-6409
George V Lauder ⓘ http://orcid.org/0000-0003-0731-286X

### Ethics

Animal holding and experimental procedures were approved by the Harvard University Animal Care IACUC Committee (protocol number 20-03-3).

Reviewer #2 (Public Review): https://doi.org/10.7554/eLife.90352.3.sa1
Reviewer #3 (Public Review): https://doi.org/10.7554/eLife.90352.3.sa2
Author Response https://doi.org/10.7554/eLife.90352.3.sa3

## Additional files

### Supplementary files

• MDAR checklist

• Supplementary file 1. We provide additional information on the materials and methods as well as supplementary tables. These materials are a supplement to the main text and explain the aspects of the methods in more detail and document how we measured both the aerobic and non-aerobic contributions to total energy expenditure (TEE) during locomotion both by individuals and by the replicate schools of giant danio. Also included in this supplemental material are tables that summarize the various terms used for fish group locomotor behaviour and literature on fish schooling swim speeds in nature. We would like to emphasize that we focus on active, directional fish schooling behaviour and that such behaviour in the wild involves fish swimming at relatively high

speeds, up to 20 body lengths second$^{-1}$. This contrasts with many previous studies of fish schooling that have been conducted in still water. When fish swim at high speeds they engage in anaerobic glycolysis, and this can contribute significantly to the total cost of swimming. In this study, we measure both the aerobic and non-aerobic costs of active directional schooling behaviour in fishes, and by studying fish groups swimming at higher speeds we demonstrate directly the substantial energy savings resulting from group locomotion.

## Data availability

All data generated or analysed during this study are included in this published article (and *Supplementary file 1*).

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

# Appendix 1

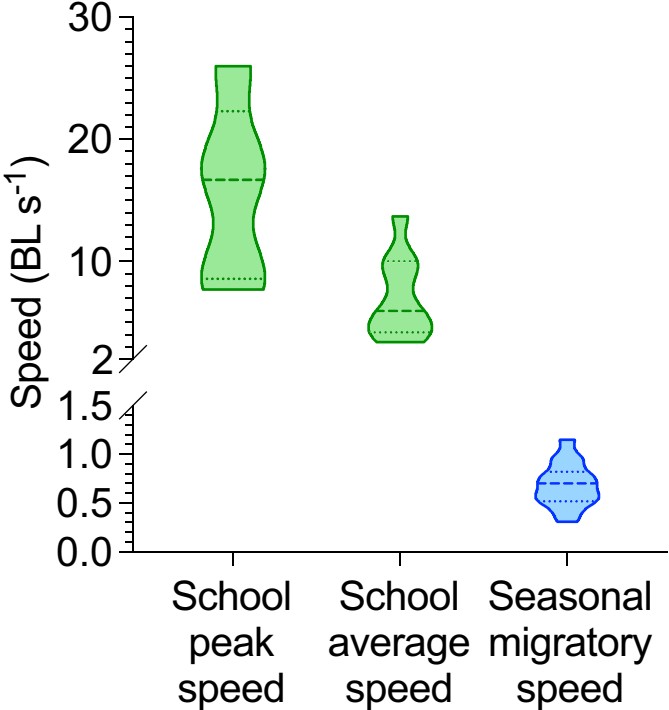

**Appendix 1—figure 1.** Swimming speeds of fish schools and seasonal migratory speeds in the open ocean. The figure is based on a summary of the swimming speed data from the literature (see *Supplementary file 1* for details). We report the relative swimming speed (in body length per second, BL s$^{-1}$) to account for the size range of species studied. We distinguished the data reported for peak speed and average speed for fish schools. Notably, schools can reach a peak speed that is nearly twice that of the average speed when they school actively in the ocean. Seasonal migratory speeds are most commonly obtained from animal-borne sensors, and data in the current literature currently skew toward large migratory species due to the technical limitations of attaching relatively large tags. Nevertheless, these are the best data available to estimate migratory speeds over long durations. A key criterion that we used in compiling these data is that the measurements are obtained from field settings e.g. estuaries and oceans. Our objective is to understand a wide range of swimming speeds of fish schools in natural habitats. This analysis demonstrates that fish schools often swim at high speeds (> 6 body length s$^{-1}$) when moving in open-water habitats. According to the results of our metabolic testing, the high speeds of swimming observed in fish schools are sustained by a substantial amount of energy contributed by glycolytic metabolism. At higher speeds, our laboratory testing results demonstrate that schooling dynamics saved a substantial amount of total energy expenditure (TEE) when the energetic cost by fluid drag is at a premium. Data on seasonal migratory speeds also match our laboratory testing results showing that solitary individuals and fish schools have a minimum absolute energetic cost when swimming around ~1 BL s$^{-1}$. A version of this figure is published in *Zhang and Lauder, 2023*.

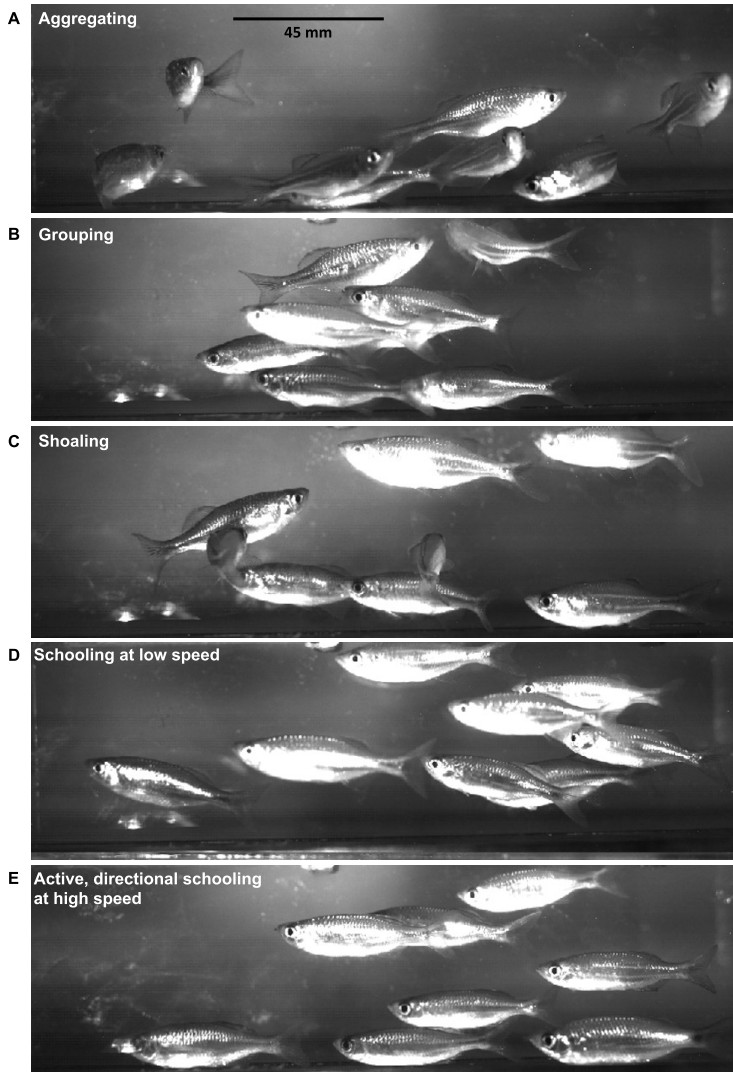

**Appendix 1—figure 2.** Frames from high-speed video to show different formations of fish schools studied in the respirometer. (**A**) Aggregating is non-directional swimming at low speeds with low net forward speed; (**B**) Grouping is a non-specific aggregation with fish more closely grouped together; (**C**) Shoaling is often manifested as individuals in groups showing various orientations with low mean forward directional group movement speed; (**D**) Active directional schooling at low speed occurs when all fish have similar body orientation with low mean group forward speed accompanied by few individuals showing a variety of inclined body angles; (**E**) Active directional schooling at high speed occurs when all fish swimming in the school have nearly the same body orientation with high mean group forward speed and all fish are oriented with body angles directed into the oncoming flow. *Supplementary file 1a* presents different terms used to describe fish collective behaviors and the swimming speeds commonly associated with each term.

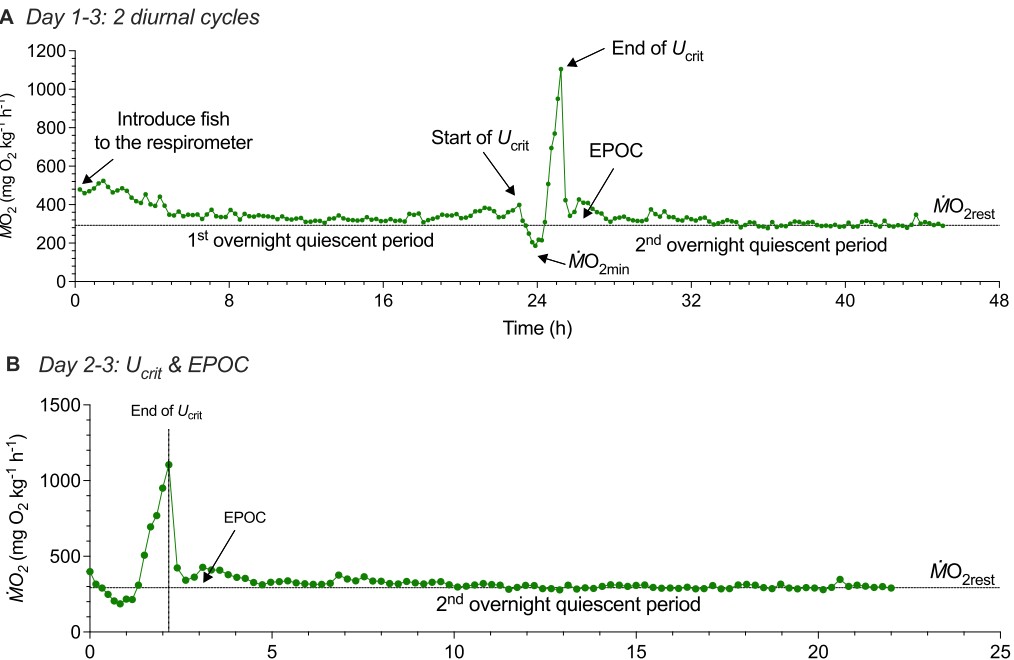

**Appendix 1—figure 3.** Three-day protocol for measuring whole-animal oxygen uptake ($\dot{M}O_2$) to estimate the contribution of aerobic and non-aerobic energy cost to the total energy expenditure of schooling locomotion. (**A**) The first ~20 h quiescent state provides a reliable baseline to estimate a resting $\dot{M}O_2$ ($\dot{M}O_{2rest}$). During this period, fish have not experienced the treatment effect, e.g., the swimming performance test on the 2nd day of the protocol. To minimize the inevitable handling stress, fish were gently placed in the swim-tunnel respirometry at the start of the protocol. The stable and flat $\dot{M}O_2$ trace during the 1st overnight quiescent period provides a reliable estimate of aerobic metabolic demand when fish are in a resting state. On the second day, a 140 min critical swimming speed ($U_{crit}$) test characterizes the active $\dot{M}O_2$ of fish as a function of water velocity to evaluate the aerobic cost of swimming performance. (**B**) After the $U_{crit}$ test, fish are left undisturbed in the respirometer for a subsequent ~20 h to measure excess post-exercise oxygen consumption (EPOC), a metric that quantifies $O_2$ cost of non-aerobic activities related to the high-energy phosphate and anaerobic glycolysis. EPOC is calculated as the integral area bounded by recovery $\dot{M}O_2$ and $\dot{M}O_{2rest}$ +10%. A majority of EPOC is the deferred anaerobic cost that supported swim performance at higher speeds. Altogether, EPOC (post-$U_{crit}$ test) and active $\dot{M}O_2$ (during the $U_{crit}$ test) measure the total energy expenditure (TEE) of *both* aerobic and non-aerobic costs. Notably, the post-exercise $\dot{M}O_{2rest}$ returned to a stable level prior to the $U_{crit}$ test. Had we not conducted the measurement of $\dot{M}O_{2rest}$ ~20 hours prior to the $U_{crit}$ test, we could not have been sure that fish have recovered to the stable baseline before the swimming performance test. Knowing the reliable baseline improves the measurement accuracy of EPOC. In addition, the entire three-day protocol is conducted under a shrouded experimental system. This achieves the objective of eliminating visual disturbances that induce spontaneous activities and impact the metabolic rate and kinematics of fish. The videos are recorded by two high-speed cameras illuminated by an array of infrared lights.

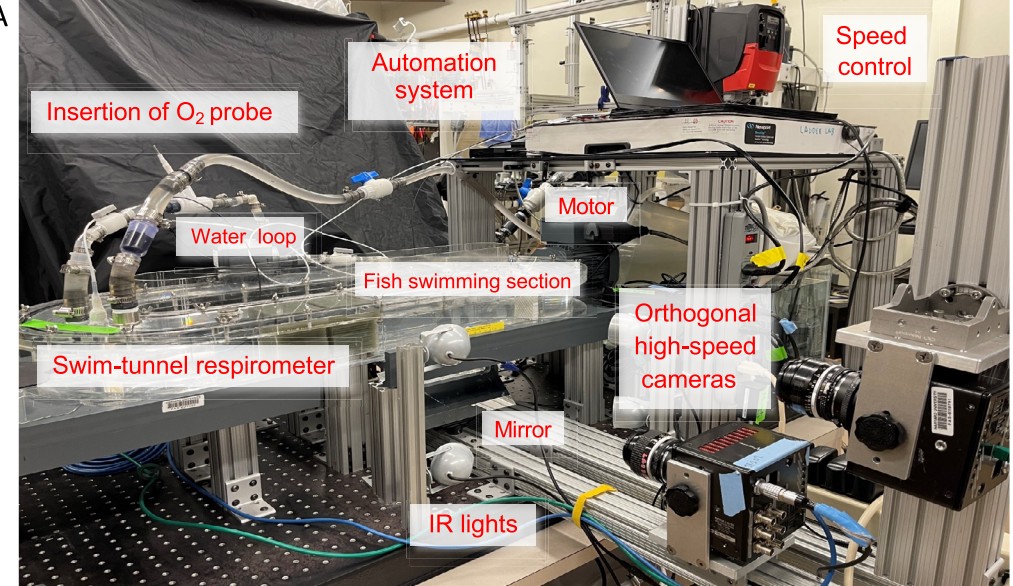

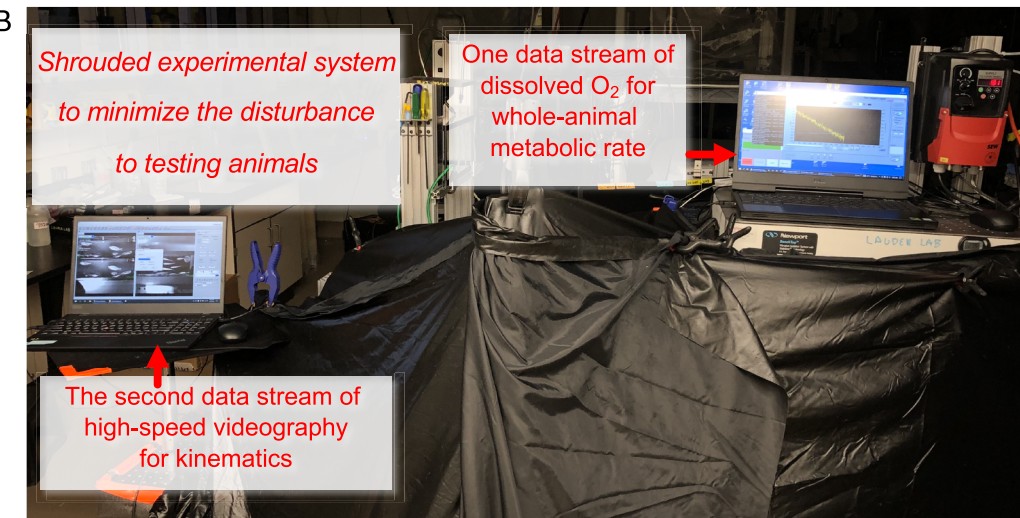

**Appendix 1—figure 4.** Experimental system of Integrated Biomechanics & Bioenergetic Assessment System (IBAS). (**A**) The important components of IBAS are labelled. The three-dimensional biomechanics measurements are obtained by two orthogonally placed high-speed cameras aimed at the fish swimming section that is illuminated by infrared (IR) lights. The swim-tunnel respirometer is equipped with an oxygen ($O_2$) probe inserted into a water loop that boosts the signal-to-noise ratio. The speed control system precisely regulates water velocity in a critical swimming speed test to measure the aerobic cost of locomotion. The respirometer is also automated by a computer system that is critical to measuring the non-aerobic cost of locomotion. (**B**) The automation system enables the fish to rest overnight in a shrouded experimental system. Minimizing the disturbance to the animals is a critical component of observing the upward concave locomotor performance curve of fish schools and solitary individuals. The automation system also measured the baseline resting metabolic rate when animals were in the 20 hr resting period before the critical swimming speed test. After the test, the automation system measured the 20 hr recovery of the whole-animal metabolic rate until it returned to the pre-testing baseline. From this, the non-aerobic cost of locomotion is measured as excess post-exercise oxygen consumption (EPOC). The IR lights enable the recording of schooling dynamics and individual kinematics when the entire system is shrouded and without bright lights influencing fish behavior. The whole-animal metabolic rate is recorded simultaneously using the respirometry system.

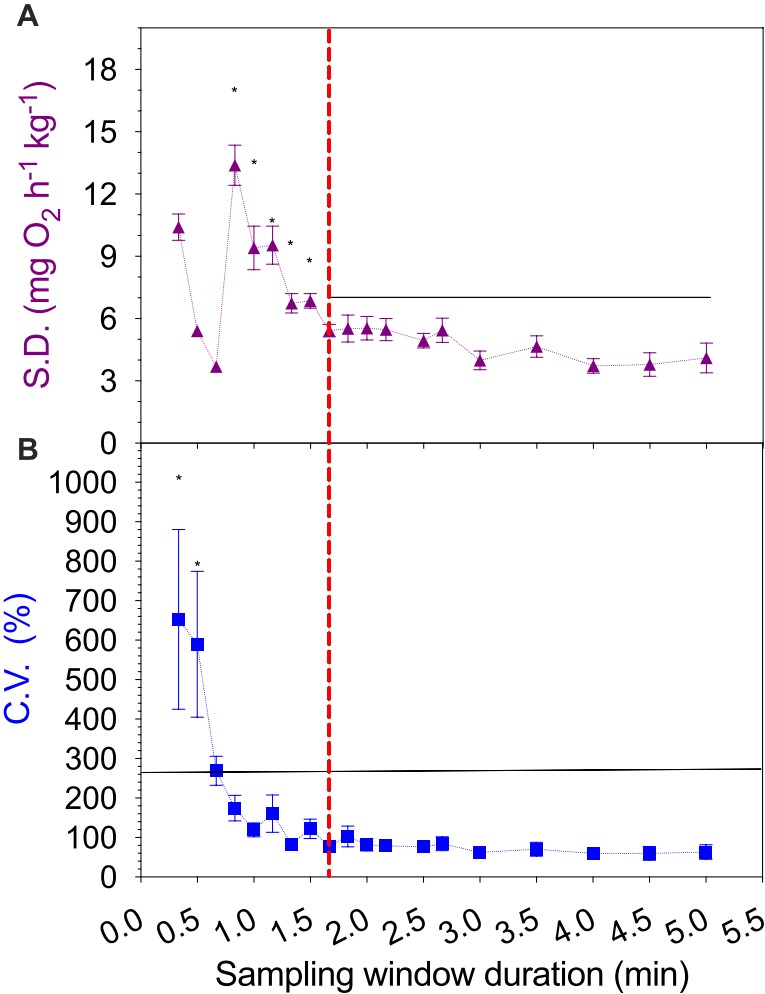

**Appendix 1—figure 5.** An analysis of the impact of varying the sampling window duration on estimates of individual rates of oxygen uptake ($\dot{M}O_2$). The signal-to-noise ratio analyses used six independent background respirometer $\dot{M}O_2$ data sets in a 20 min duration. We calculated (**A**) standard deviations (S.D.) and (**B**) coefficient of variation (C.V.) for each complete data set as a function of sampling window duration (0.8–5.0 min). The S.D. and C.V. values were compared across different sampling window durations using one-way ANOVA with Tukey *post-hoc* tests ($\alpha < 0.05$). Values are presented as mean ± s.e.m. This analysis suggests that 1.67 min as a minimum conservative sampling duration (the red vertical dashed line) using the criterion of S.D. and C.V. being stable after high variation at shorter duration windows. This suggests that the 8 min sampling duration used in this study is sufficient to resolve the measurement of steady-state $\dot{M}O_2$ for sustained swimming. The analytic method is published in *Zhang et al., 2019*.

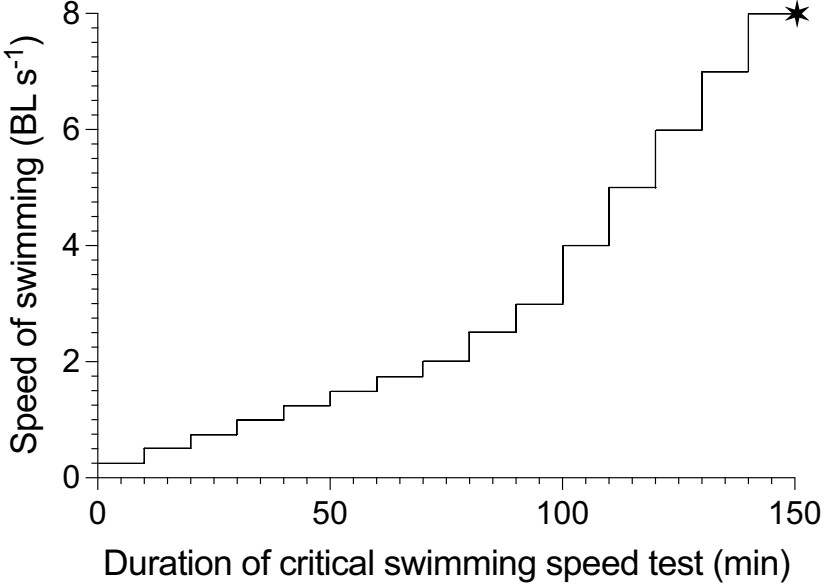

**Appendix 1—figure 6.** A schematic illustration of the incremental steps in swimming speed used for the critical swimming speed test ($U_{crit}$) protocol. Speed is presented as the relative swimming speed of fish normalized to body lengths per second (BL s$^{-1}$). Each speed increment has a 10 min duration. Each vertical increment marks the increase in speed for each step. The mean velocity at fatigue is indicated by the ✳ symbol, marking the total duration of the test, after which EPOC is measured for a 20 hr period.

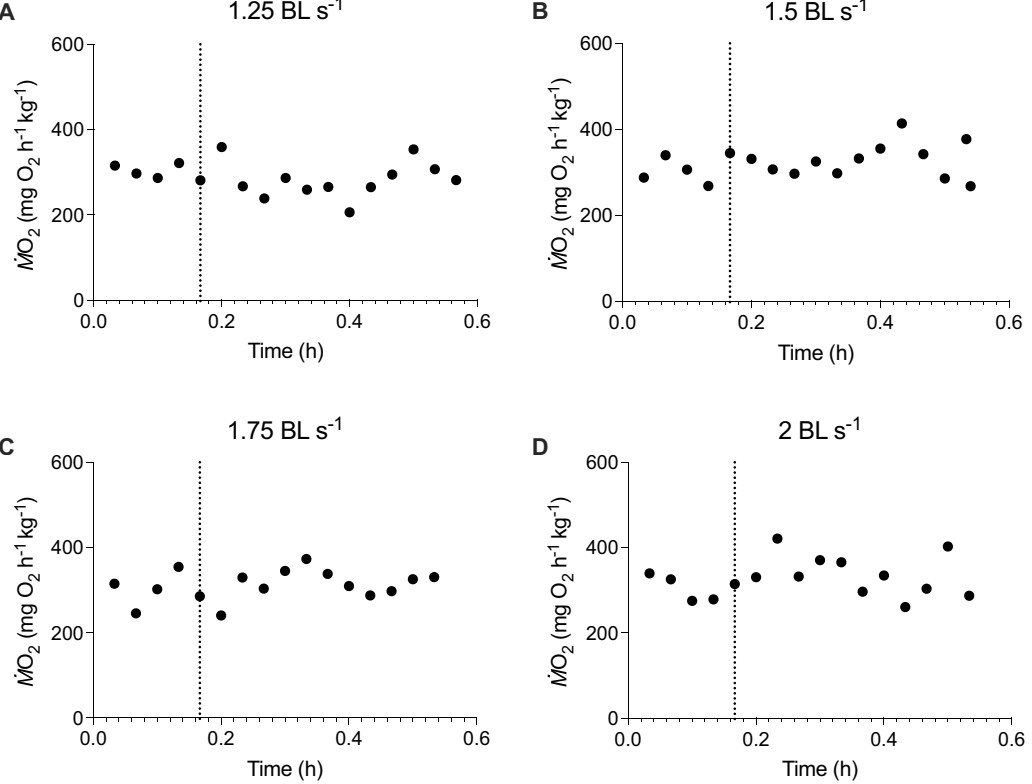

**Appendix 1—figure 7.** Stability of whole-animal oxygen uptake ($\dot{M}O_2$) profile when fish swim steadily at low water velocity. The quality of $\dot{M}O_2$ measurement over the range of relative swimming speeds (body length per sec, *Appendix 1—figure 7 continued on next page*

*Appendix 1—figure 7 continued*

BL s⁻¹) where fish show the lowest $\dot{M}O_2$ is critical to assure the accuracy of quantifying concave upward shaped metabolism-speed curve. Hence, we conducted an additional quality assurance test to inspect the stability of $\dot{M}O_2$ at 1.25 (**A**), 1.5 (**B**), 1.75 (**C**) & 2.0 (**D**) BL s⁻¹ over a 30 min period. $\dot{M}O_2$ recorded in the first 10 mins is within the same range as the ensuing 20 mins. Thus, our 10 min testing period at each water velocity provides a reliable estimate of the aerobic cost when fish swim at a given swimming speed. This conclusion is in agreement with the same testing period used in a previous study (**Di Santo et al., 2017**).

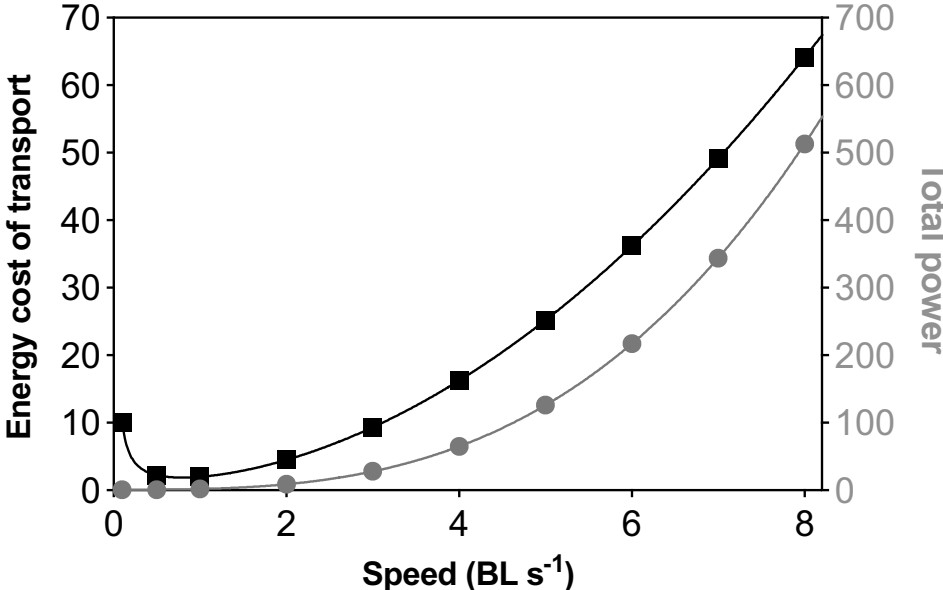

**Appendix 1—figure 8.** Illustration of the theoretical relationship between the cost of transport and total power as a function of nominal speed based on the estimated power required to swim as drag forces increase at increasing swimming speeds. The equation for the energy cost of transport is $y=0.9996\,x^{-1} + x^2$ (power series model: $R^2=1$, AIC = –127.6). The equation of total power is $y=1.001–0.000643\,x+8.564e^{-005}x^2 + 1x^3$ (polynomial model: $R^2=1$, AIC = –138.7). Notably, the same models provided best-fit equations to describe the measurement of the cost of transport and total energy expenditure (aerobic and glycolytic metabolism) on live fish. This suggests that total energy expenditure during fish locomotion at increasing speed may be largely due to the requirement to overcome fluid dynamic drag. The theoretical numbers (nominal) of power and energy cost of transport are adopted from D. Robinson, Ed., Animal Performance (The Open University, Milton Keynes, 1997). Speed is given in body lengths per second (BL s⁻¹) while the power and energy axes are unitless relative numbers.

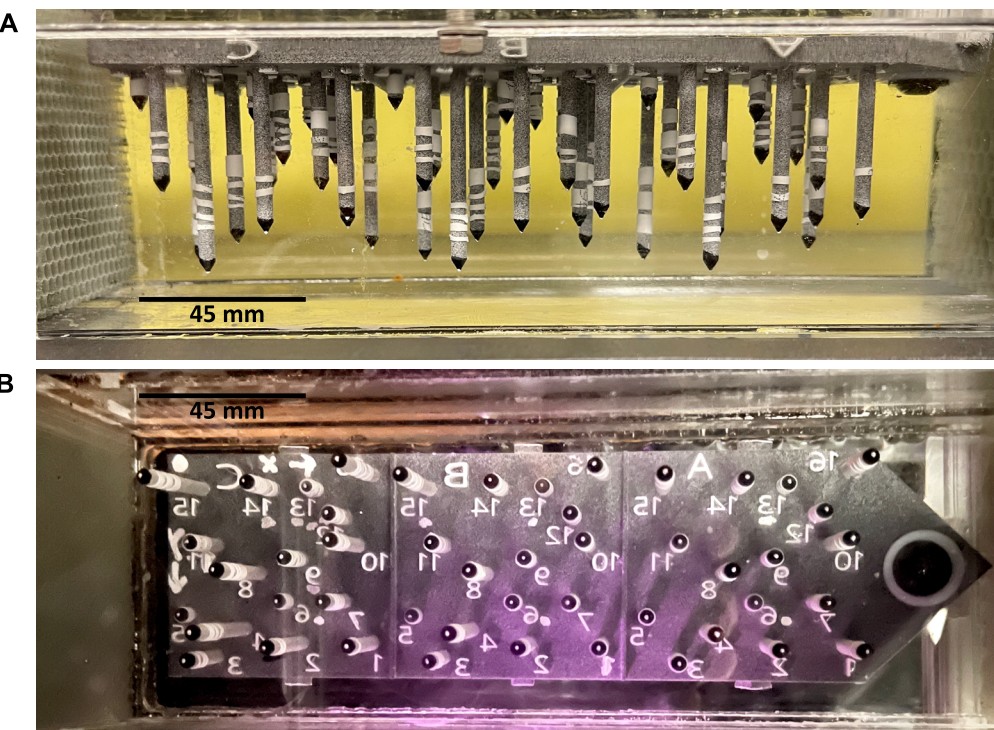

**Appendix 1—figure 9.** Photographs of the 'spike' calibration device used to map the three-dimensional (3-D) space of the fish swimming section inside the swim-tunnel respirometer. This direct linear transformation calibration system provided accurate 3D coordinates within the swimming arena for both individual danio and schools of up to 8 danio. The tip of each spike provides a marker for spatial coordinates. Each tip has a known three-dimensional coordinate based on the known distance on the x-(length), y-(width) & z-(height) axes. Lateral (**A**) and dorsal (**B**) views are provided. The 'spike' calibration provides 45 spatial points to sufficiently cover a volume of $1440 \cdot 10^3$ mm$^3$. The spatial calibration is used by DLTdv8 software (*Hedrick, 2008*, Bioinspiration & Biomimetics 3, 034001) to extract the three-dimensional coordinates for locations digitized on individual fish (see *Appendix 1—figure 10*).

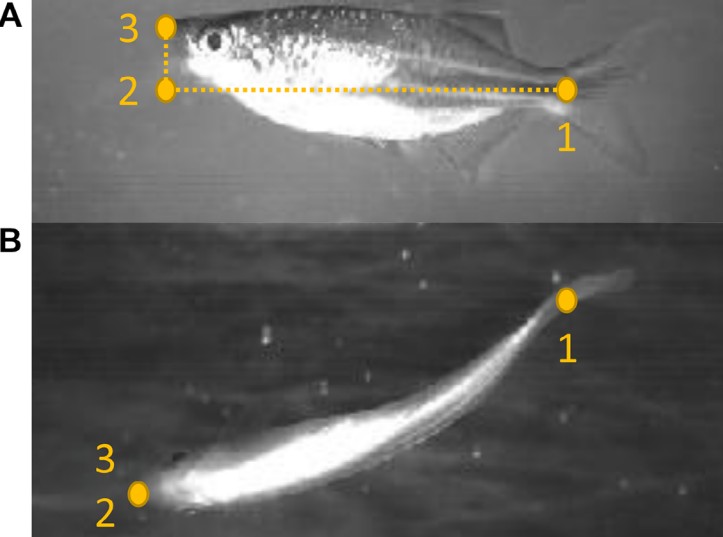

**Appendix 1—figure 10.** Landmarks placed on individual giant danio (*Devario aequipinnatus*) as well as individuals within the school to digitize 3D body locations during locomotion. We marked the peduncle (marker #1), upper jaw (marker #3) and the right angle cross point (marker #2) of the peduncle and jaws for each individual (both for *Appendix 1—figure 10 continued on next page*

**Appendix 1—figure 10 continued**

solitary individuals and fish in schools). The markers are placed on both lateral (**A**) and ventral (**B**) views. The 3rd marker allows us to estimate two vectors for each fish, which enables the calculation of body angles in both the lateral and ventral views.

