## [Editor Report · eLife assessment]

The authors provide an **important** series of metabolic measurements characterizing group dynamics in fish, rationalizing that schooling behavior presents several benefits. The strength of evidence supporting this conclusion is **solid**, but the specific methodological and analytical approaches taken should be considered for further interpretation.

---

## [Referee Report · Reviewer #2 (Public Review)]

Summary:

This paper tests the idea that schooling can provide an energetic advantage over solitary swimming. The present study measures oxygen consumption over a wide range of speeds, to determine the differences in aerobic and anaerobic cost of swimming, providing a potentially valuable addition to the literature related to the advantages of group living.

Strengths:

The strength of this paper is related to providing direct measurements of the energetics (oxygen consumption) of fish while swimming in a group vs solitary. The energetic advantages of schooling has been claimed to be one of the major advantages of schooling and therefore a direct energetic assessment is a useful result.

Weaknesses:

1. Regarding the fish to water volume ratio, the arguments raised by the authors are valid. However, the ratio used is still quite high (as high as >2000 in solitary fish), much higher than that recommended by Svendsen et al (2006). Hence this point needs to be discussed in the ms (summarising the points raised in the authors' response)

2. Wall effects: Fish in a school may have been swimming closer to the wall. The fact that the convex hull volume of the fish school did not change as speed increased is not a demonstration that fish were not closer to the wall, nor is it a demonstration that wall effect were not present. Therefore the issue of potential wall effects is a weakness of this paper.

3. The authors stated "Because we took high-speed videos simultaneously with the respirometry measurements, we can state unequivocally that individual fish within the school did not swim closer to the walls than solitary fish over the testing period". This is however not quantified.

4. Statistical analysis. The authors have dealt satisfactorily with most of the comments.

---

## [Referee Report · Reviewer #3 (Public Review)]

Zhang and Lauder characterized both aerobic and anaerobic metabolic energy contributions in schools and solitary fishes in the Giant danio (Devario aequipinnatus) over a wide range of water velocities. By using a highly sophisticated respirometer system, the authors measure the aerobic metabolisms by oxygen uptake rate and the non-aerobic oxygen cost as excess post-exercise oxygen consumption (EPOC). With these data, the authors model the bioenergetic cost of schools and solitary fishes. The authors found that fish schools have a J-shaped metabolism-speed curve, with reduced total energy expenditure per tail beat compared to solitary fish. Fish in schools also recovered from exercise faster than solitary fish. Finally, the authors conclude that these energetic savings may underlie the prevalence of coordinated group locomotion in fish.

The conclusions of this paper are mostly well supported by data.

---

## [Author Response]

The following is the authors’ response to the current reviews.

**Public Reviews:**

**Reviewer #2 (Public Review):**
Summary:This paper tests the idea that schooling can provide an energetic advantage over solitary swimming. The present study measures oxygen consumption over a wide range of speeds, to determine the differences in aerobic and anaerobic cost of swimming, providing a potentially valuable addition to the literature related to the advantages of group living.

Response: Thank you for the positive comments.

Strengths:The strength of this paper is related to providing direct measurements of the energetics (oxygen consumption) of fish while swimming in a group vs solitary. The energetic advantages of schooling has been claimed to be one of the major advantages of schooling and therefore a direct energetic assessment is a useful result.

Response: Thank you for the positive comments.

Weaknesses:1. Regarding the fish to water volume ratio, the arguments raised by the authors are valid. However, the ratio used is still quite high (as high as >2000 in solitary fish), much higher than that recommended by Svendsen et al (2006). Hence this point needs to be discussed in the ms (summarising the points raised in the authors' response)

Response: Thank you for the comments. We have addressed this point in the previous comments. In short, our ratio is within the range of the published literature. We conducted the additional signal-to-noise analysis for quality assurance.

1. Wall effects: Fish in a school may have been swimming closer to the wall. The fact that the convex hull volume of the fish school did not change as speed increased is not a demonstration that fish were not closer to the wall, nor is it a demonstration that wall effect were not present. Therefore the issue of potential wall effects is a weakness of this paper.

Response: Thank you for the comments. We have addressed this point in the previous comments. We provided many other considerations in addition to the convex hull volume. In particular, our boundary layer is < 2.5mm, which was narrower than the width of the giant danio of ~10 mm.

1. The authors stated "Because we took high-speed videos simultaneously with the respirometry measurements, we can state unequivocally that individual fish within the school did not swim closer to the walls than solitary fish over the testing period". This is however not quantified.

Response: Thank you for the comments. We have addressed this point in the previous comments. We want to note that the statement in the response letter is to elaborate the discussion points, but not stated as data in the manuscript. The bottom line is very few studies used PIV to quantify the thickness of the boundary layer like what we did in our experiment.

1. Statistical analysis. The authors have dealt satisfactorily with most of the comments.However :(a) the following comment has not been dealt with directly in the ms "One can see from the graphs that schooling MO2 tends to have a smaller SD than solitary data. This may well be due to the fact that schooling data are based on 5 points (five schools) and each point is the result of the MO2 of five fish, thereby reducing the variability compared to solitary fish."(b) Different sizes were used for solitary and schooling fishes. The authors justify using larger fish as solitary to provide a better ratio of respirometer volume to fish volume in the tests on individual fish. However, mass scaling for tail beat frequency was not provided. Although (1) this is because of lack of data for this species and (2) using scaling exponent of distant species would introduce errors of unknown magnitude, this is still a weakness of the paper that needs to be acknowledged here and in the ms.

Response: Thank you for the comments. We have addressed both points in the previous comments and provided comprehensive discussions. We also stated the caveats in the method section of the manuscript.

**Reviewer #3 (Public Review):**
Zhang and Lauder characterized both aerobic and anaerobic metabolic energy contributions in schools and solitary fishes in the Giant danio (Devario aequipinnatus) over a wide range of water velocities. By using a highly sophisticated respirometer system, the authors measure the aerobic metabolisms by oxygen uptake rate and the non-aerobic oxygen cost as excess post-exercise oxygen consumption (EPOC). With these data, the authors model the bioenergetic cost of schools and solitary fishes. The authors found that fish schools have a J-shaped metabolism-speed curve, with reduced total energy expenditure per tail beat compared to solitary fish. Fish in schools also recovered from exercise faster than solitary fish. Finally, the authors conclude that these energetic savings may underlie the prevalence of coordinated group locomotion in fish.The conclusions of this paper are mostly well supported by data.

Response: Thank you for the positive comments.

**Recommendations for the authors:**

**Reviewer #3 (Recommendations For The Authors):**
I have read carefully the revised version of the manuscript and would like to thank the authors for addressing all my comments/suggestions.I have no additional comments/suggestions. Now, I strongly believe that this manuscript deserves to be published in eLife.

Response: Thank you for the positive comments.

The following is the authors’ response to the original reviews.

General responses

Many thanks to the reviewers and editors for their very helpful comments on our manuscript. Below we respond (in blue text) to each of the reviewer comments, both the public ones and the more detailed individual comments in the second part of each review. In some cases, we consider these together where the same point is made in both sets of comments. We have made several changes to the manuscript in response to reviewer suggestions, and we respond in detail to the comments of reviewer #2 who feels that we have overstated the significance of our manuscript and suggests several relevant literature references. We prepared a table summarizing these references and why they differ substantially from the approach taken in our paper here.

Overall, we would like to emphasize to both reviewers and readers of this response document that previous studies of fish schooling dynamics (or collective movement of vertebrates in general, see Commentary Zhang & Lauder 2023 J. Exp. Biol., doi:10.1242/jeb.245617) have not considered a wide speed range and thus the importance of measuring EPOC (excess post-exercise oxygen consumption) as a key component of energy use. Quantifying both aerobic and non-aerobic energy use allows us to calculate the total energy expenditure (TEE) which we show differs substantially and, importantly, non-linearly with speed between schools and measurements on solitary individuals. Comparison between school total energy use and individual total energy use are critical to understanding the dynamics of schooling behaviour in fishes.

The scope of this study is the energetics of fish schools. By quantifying the TEE over a wide range of swimming speeds, we also show that the energetic performance curve is concave upward, and not linear, and how schooling behaviour modifies this non-linear relationship.

In addition, one key implication of our results is that kinematic measurements of fish in schools (such as tail beat frequency) are not a reliable metric by which to estimate energy use. Since we recorded high-speed video simultaneously with energetic measurements, we are able to show that substantial energy savings occur by fish in schools with little to no change in tail beat frequency, and we discuss in the manuscript the various fluid dynamic mechanisms that allow this. Indeed, studies of bird flight show that when flying in a (presumed) energy-saving V-formation, wing beat frequency can actually increase compared to flying alone. We believe that this is a particularly important part of our findings: understanding energy use by fish schools must involve actual measurements of energy use and not indirect and sometimes unreliable kinematic measurements such as tail beat frequency or amplitude.

**Reviewer #1 (Public Review):**
Summary:In the presented manuscript the authors aim at quantifying the costs of locomotion in schooling versus solitary fish across a considerable range of speeds. Specifically, they quantify the possible reduction in the cost of locomotion in fish due to schooling behavior. The main novelty appears to be the direct measurement of absolute swimming costs and total energy expenditure, including the anaerobic costs at higher swimming speeds.In addition to metabolic parameters, the authors also recorded some basic kinematic parameters such as average distances or school elongation. They find both for solitary and schooling fish, similar optimal swimming speeds of around 1BL/s, and a significant reduction in costs of locomotion due to schooling at high speeds, in particular at ~5-8 BL/s.Given the lack of experimental data and the direct measurements across a wide range of speeds comparing solitary and schooling fish, this appears indeed like a potentially important contribution of interest to a broader audience beyond the specific field of fish physiology, in particular for researchers working broadly on collective (fish) behavior.

Response: Thank you for seeing the potential implications of this study. We also believe that this paper has broader implications for collective behaviour in general, and outline some of our thinking on this topic in a recent Commentary article in the Journal of Experimental Biology: (Zhang & Lauder 2023 doi:10.1242/jeb.245617). Understanding the energetics of collective behaviours in the water, land, and air is a topic that has not received much attention despite the widespread view that moving as a collective saves energy.

Strengths:The manuscript is for the most part well written, and the figures are of good quality. The experimental method and protocols are very thorough and of high quality. The results are quite compelling and interesting. What is particularly interesting, in light of previous literature on the topic, is that the authors conclude that based on their results, specific fixed relative positions or kinematic features (tail beat phase locking) do not seem to be required for energetic savings. They also provide a review of potential different mechanisms that could play a role in the energetic savings.

Response: Thank you for seeing the nuances we bring to the existing literature and comment on the quality of the experimental method and protocols. Despite a relatively large literature on fish schooling based on previous biomechanical research, our studies suggest that direct measurement of energetic cost clearly demonstrates the energy savings that result from the sum of different fluid dynamic mechanisms depending on where fish are, and also emphasizes that simple metrics like fish tail beat frequency do not adequately reflect energy savings during collective motion.

Weaknesses:A weakness is the actual lack of critical discussion of the different mechanisms as well as the discussion on the conjecture that relative positions and kinematic features do not matter. I found the overall discussion on this rather unsatisfactory, lacking some critical reflections as well as different relevant statements or explanations being scattered across the discussion section. Here I would suggest a revision of the discussion section.

Response: The critical discussion of the different possible energy-saving mechanisms is indeed an important topic. We provided a discussion about the overall mechanism of ‘local interactions’ in the first paragraph of “Schooling Dynamics and energy conservation”. To clarify, our aim with Figure 1 is to introduce the current mechanisms proposed in the existing engineering/hydrodynamic literature that have studied a number of possible configurations both experimentally and computationally. Thank you for the suggestion of better organizing the discussion to critically highlight different mechanisms that would enable a dynamic schooling structure to still save energy and why the appendage movement frequency does not necessarily couple with the metabolic energy expenditure. Much of this literature uses computational fluid dynamic models or experiments on flapping foils as representative of fish. This exact issue is of great interest to us, and we are currently engaged in a number of other experiments that we hope will shed light on how fish moving in specific formations do or don’t save energy.

Our aim in presenting Figure 1 at the start of the paper was to show that there are several ways that fish could save energy when moving in a group as shown by engineering analyses, but before investigating these various mechanisms in detail we first have to show that fish moving in groups actually do save energy with direct metabolic measurements. Hence, our paper treats the various mechanisms as inspiration to determine experimentally if, in fact, fish in schools save energy, and if so how much over a wide speed range. Our focus is to experimentally determine the performance curve that shows energy use as speed increases, for schools compared to individuals. Therefore, we have elected not to go into detail about these different hydrodynamic mechanisms in this paper, but rather to present them as a summary of current engineering literature views and then proceed to document energy savings (as stated in the second last paragraph of Introduction). We have an Commentary paper in the Journal of Experimental Biology that addresses this issue generally, and we are reluctant to duplicate much of that discussion here (Zhang & Lauder 2023 doi:10.1242/jeb.245617). We are working hard on this general issue as we agree that it is very interesting. We have revised the Introduction (second last paragraph of Introduction) and Discussion (first paragraph of Discussion) to better indicate our approach, but we have not added any significant discussion of the different hydrodynamic energy saving proposals as we believe that it outside the scope of this first paper and more suitable as part of follow-up studies.

Also, there is a statement that Danio regularly move within the school and do not maintain inter-individual positions. However, there is no quantitative data shown supporting this statement, quantifying the time scales of neighbor switches. This should be addressed as core conclusions appear to rest on this statement and the authors have 3d tracks of the fish.

Response: Thank you for pointing out this very important future research direction. Based on our observations and the hypothesized mechanisms for fish within the school to save energy (Fig. 1), we have been conducting follow-up experiments to decipher the multiple dynamic mechanisms that enable the fish within the school to save energy. Tracking the 3D position of each individual fish body in 3D within the fish school has proven difficult. We currently have 3D data on the nose position obtained simultaneously with the energetic measurements, but we do not have full 3D fish body positional data. Working with our collaborators, we are developing a 3-D tracking algorithm that will allow us to quantify how long fish spend in specific formations, and we currently have a new capability to record high-speed video of fish schooling moving in a flow tank for many hours (see our recent perspective by Ko et al., 2023 doi.org/10.1098/rsif.2023.0357). The new algorithms and the results will be published as separate studies and we think that these ongoing experiments are outside the scope of the current study with its focus on energetics. Nevertheless, the main point of Fig. 1 is to provide possible mechanisms to inspire future studies to dissect the detailed hydrodynamic mechanisms for energy saving, and the points raised by this comment are indeed extremely interesting to us and our ongoing experiments in this area. We provide a statement to clarify this point in the 1st paragraph of “Schooling dynamics and energy conservation” section.

Further, there is a fundamental question on the comparison of schooling in a flow (like a stream or here flow channel) versus schooling in still water. While it is clear that from a pure physics point of view that the situation for individual fish is equivalent. As it is about maintaining a certain relative velocity to the fluid, I do think that it makes a huge qualitative difference from a biological point of view in the context of collective swimming. In a flow, individual fish have to align with the external flow to ensure that they remain stationary and do not fall back, which then leads to highly polarized schools. However, this high polarization is induced also for completely non-interacting fish. At high speeds, also the capability of individuals to control their relative position in the school is likely very restricted, simply by being forced to put most of their afford into maintaining a stationary position in the flow. This appears to me fundamentally different from schooling in still water, where the alignment (high polarization) has to come purely from social interactions. Here, relative positioning with respect to others is much more controlled by the movement decisions of individuals. Thus, I see clearly how this work is relevant for natural behavior in flows and that it provides some insights on the fundamental physiology, but I at least have some doubts about how far it extends actually to “voluntary” highly ordered schooling under still water conditions. Here, I would wish at least some more critical reflection and or explanation.

Response: We agree completely with this comment that animal group orientations in still fluid can have different causes from their locomotion in a moving fluid. We very much agree with the reviewer that social interactions in still water, which typically involve low-speed locomotion and other behaviours such as searching for food by the group, can be important and could dictate fish movement patterns. In undertaking this project, we wanted to challenge fish to move at speed, and reasoned that if energy savings are important in schooling behaviour due to hydrodynamic mechanisms, we should see this when fish are moving forward against drag forces induced by fluid impacting the school. Drag forces scale as velocity squared, so we should see energy savings by the school, if any, as speed increases.

We also quantified fish school swimming speeds in the field from the literature and presented a figure showing that in nature fish schools can and do move at considerable speeds. This figure is part of our overview on collective behaviour recently in J. Exp. Biol. (Zhang & Lauder 2023 doi:10.1242/jeb.245617). It is only by studying fish schools moving over a speed range that we can understand the performance curve relating energy use to swimming speed. Indeed, we wonder if fish moving in still water as a collective versus as solitary individuals would show energy savings at all.We now provided the justification for studying fish schooling in moving fluids in the second and third paragraph of the Introduction. When animals are challenged hydrodynamically (e.g. at higher speed), it introduces the need to save energy. Movement in still water lacks the need for fish to save energy. When fish do not need to save locomotor energy in still water, it is hard to justify why we would expect to observe energy saving and related physiological mechanisms in the first place. As the reviewer said, the ‘high polarization in still water has to come purely from social interactions’. Our study does not dispute this consideration, and indeed we agree with it! In our supplementary materials, we acknowledged the definitions for different scenarios of fish schooling can have different behavioural and ecological drivers. Using these definitions, we explicitly stated, in the introduction, that our study focuses on active and directional schooling behaviour to understand the possible hydrodynamic benefits of energy expenditure for collective movements of fish schools. By stating the scope of our study at the outset, we hope that this will keep the discussion focused on the energetics and kinematics of fish schools, without unnecessarily addressing other many possible reasons for fish schooling behaviours in the discussion such as anti-predator grouping, food searching, or reproduction as three examples.

As this being said, we acknowledge (in the 2nd paragraph of the introduction) that fish schooling behaviour can have other drivers when the flow is not challenging. Also, there are robotic-&-animal interaction studies and computational fluid dynamic simulation studies (that we cited) that show individuals in fish schools interact hydrodynamically. Hydrodynamic interactions are not the same as behaviour interactions, but it does not mean individuals within the fish schooling in moving flow are not interacting and coordinating.

Related to this, the reported increase in the elongation of the school at a higher speed could have also different explanations. The authors speculate briefly it could be related to the optimal structure of the school, but it could be simply inter-individual performance differences, with slower individuals simply falling back with respect to faster ones. Did the authors test for certain fish being predominantly at the front or back? Did they test for individual swimming performance before testing them in groups together? Again this should be at least critically reflected somewhere.

Response: Thank you for raising this point. If the more streamlined schooling structure above 2 BL/s is due to the weaker individuals not catching up with the rest of the school, we would expect the weaker individuals to quit swimming tests well before 8 BL/s. However, we did not observe this phenomenon. Although we did not specifically test for the two questions the reviewer raises here, our results suggest that inter-individual variation in the swimming performance of giant Danio is not at the range of 2 to 8 BL/s (a 400% difference). While inter-individual differences certainly exist, we believe that they are small relative to the speeds tested as we did not see any particular individuals consistently unable to keep up with the school or certain individuals maintaining a position near the back of the school. As this being said, we provide additional interpretations for the elongated schooling structure at the end of the 2nd paragraph of the “schooling dynamics and energy conservation” section.

**Reviewer #1 (Recommendations For The Authors):**
Line 58: The authors write "How the fluid dynamics (...) enable energetic savings (...)". However, the paper focuses rather on the question of whether energetic savings exist and does not enlighten us on the dominant mechanisms. Although it gives a brief overview of all possible mechanisms, it remains speculative on the actual fluid dynamical and biomechanical processes. Thus, I suggest changing "How" to "Whether".

Response: Great point! We changed “How” to “Whether”.

Lines 129-140: In the discussion of the U-shaped aerobic rate, there is no direct comparison of the minimum cost values between the schooling and solitary conditions. Only the minimum costs during schooling are named/discussed. In addition to the data in the figure, I suggest explicitly comparing them as well for full transparency.

Response: Thanks for raising this point. We did not belabor this point because there was no statistical significance. As requested, we added a statement to address this with statistics in the 1st paragraph of the Results section.

Line 149: The authors note that the schooling fish have a higher turning frequency than solitary fish. Here, a brief discussion of potential explanations would be good, e.g. need for coordination with neighbors -> cost of schooling.

Response: Thank you for the suggestion. In the original version of the manuscript, we discussed that the higher turning frequency could be related to higher postural costs for active stability adjustment at low speeds. As requested, we now added that high turn frequency can relate to the need for coordination with neighbours in the last paragraph of the “Aerobic metabolic rate–speed curve of fish schools” section. As indicated above, the suspected costs of coordination did not result in higher costs of schooling at the lower speed (< 2 BL s-1, where the turn frequency is higher).

Line 151: The authors discuss the higher maximum metabolic rate of schooling fish as a higher aerobic performance and lower use of aerobic capacity. This may be confusing for non-experts in animal physiology and energetics of locomotion. I recommend providing somewhere in a paper an additional explanation to clarify it to non-experts. While lines 234-240 and further below potentially address this, I found this not very focused or accessible to non-experts. Here, I suggest the authors consider revisions to make it more comprehensible to a wider, interdisciplinary audience.

Response: We agree with the reviewer that the difference between maximum oxygen uptake and maximum metabolic rate can be confusing. In fact, among animal physiologists, these two concepts are often muddled. One of the authors is working on an invited commentary from J. Exp. Biol. to clearly define these two concepts. We have made the language in the section “Schooling dynamics enhances aerobic performance and reduces non-aerobic energy use” more accessible to a general audience. In addition, the original version presented the relevant framework in the first and the second paragraphs of the Introduction when discussing aerobic and non-aerobic energy contribution. In brief, when vertebrates exhibit maximum oxygen uptake, they use aerobic and non-aerobic energy contributions that both contribute to their metabolic rate. Therefore, the maximum total metabolic rate is higher than the one estimated from only maximum oxygen uptake. We used the method presented in Fig. 3a to estimate the maximum metabolic rate for metabolic energy use (combining aerobic and non-aerobic energy use). In kinesiology, maximum oxygen uptake is used to evaluate the aerobic performance and energy use of human athletes is estimated by power meters or doubly labelled water.

Line 211: The authors write that Danio regularly move within the school and do not maintain inter-individual positions. Given that this is an important observation, and the relative position and its changes are crucial to understanding the possible mechanisms for energetic savings in schools, I would expect some more quantitative support for this statement, in particular as the authors have access to 3d tracking data. For example introducing some simple metrics like average time intervals between swaps of nearest neighbors, possibly also resolved in directions (front+back versus right+left), should provide at least some rough quantification of the involved timescales, whether it is seconds, tens of seconds, or minutes.

Response: As responded in the comment above, 3-D tracking of both body position and body deformation of multiple individuals in a school is not a trivial research challenge and we have ongoing research on this issue. We hope to have results on the 3D positions of fish in schools soon! For this manuscript, we believe that the data in Figure 4E which shows the turning frequency of fish in schools and solitary controls shows the general phenomenon of fish moving around (as fish turn to change positions within the school), but we agree that more could be done to address this point and we are indeed working on it now.

Lines 212-217: There is a very strong statement that energetic savings by collective motion do not require fixed positional arrangements or specific kinematic features. While possibly one of the most interesting findings of the paper, I found that in its current state, it was not sufficiently/satisfactorily discussed. For example for the different mechanisms summarized, there will be clearly differences in their relevance based on relative distance and position. For example mechanisms 3 and 4 likely have significant contributions only at short distances. Here, the question is how relevant can they be if the average distance is 1 BL? Also, 1BL side by side is very much different from 1BL front to back, given the elongated body shape. For mechanisms 1 and 2, it appears relative positioning is quite important. Here, having maybe at least some information from the literature (if available) on the range of wall or push effects or the required precision in relative positioning for having a significant benefit would be very much desired. Also, do the authors suggest that (a) these different effects overlap giving any position in the school a benefit, or (b) that there are specific positions giving benefits due to different mechanisms and that fish "on purpose" switch only between these energetic "sweet" spots, I guess this what is towards the end referred to as Lighthill conjecture? Given the small group size I find (a) rather unlikely, while (b) actually also leads to a coordination problem if every fish is looking for a sweet spot. Overall, a related question is whether the authors observed a systematic change in leading individuals, which likely have no, or very small, hydrodynamic benefits.

Response: Thank you for the excellent discussion on this point. As we responded above, we have softened the tone of the statement. In the original version, we were clear that the known mechanisms as summarized in Fig. 1 lead us to ‘expect’ that fish do not need to be in a fixed position to save energy.

In general, current engineering/hydrodynamic studies suggest that any fish positioned within one body length (both upstream and downstream and side by side) will benefit from one or more of the hydrodynamic mechanisms that we expect will reduce energy costs, relative to a solitary individual. Our own studies using robotic systems suggest that a leading fish will experience an added mass “push” from a follower when the follower is located within roughly ½ body length behind the leader. We cited a Computational Fluid Dynamic (CFD) study about the relative distance among individuals for energy saving to be in effect. Please keep in mind that CFD simulation is a simplified model of the actual locomotion of fish and involves many assumptions and currently only resolves the time scale of seconds (see commentary of Zhang & Lauder 2023 doi:10.1242/jeb.245617 in J. Exp. Biol. for the current challenges of CFD simulation). To really understand the dynamic positions of fish within the school, we will need 3-D tracking of fish schools with tools that are currently being developed. Ideally, we would also have simultaneous energetic measurements, but of course, this is enormously challenging and it is not clear at this time how to accomplish this.

We certainly agree that the relative positions of fish (vertically staggered or in-line swimming) do affect the specific hydrodynamic mechanisms being used. We cited the study that discussed this, but the relative positions of fish remain an active area of research. More studies will be out next few years to provide more insight into the effects of the relative positions of fish in energy saving.The Lighthill conjecture is observed in flapping foils and whether fish schools use the Lighthill conjecture for energy saving is an active area of research but still unclear. We also provided a citation about the implication of the Lighthill conjecture on fish schools. Hence, our original version stated ‘The exact energetic mechanisms….would benefit from more in-depth studies’. We agree with the reviewer that not all fish can benefit Lighthill conjecture (if fish schools use it) at any given time point, hence the fish might need to rotate in using the Lighthill conjecture. This is one more explanation for the dynamic positioning of fish in a school.

Overall, in response to the question raised, we do not believe that fish are actively searching for “sweet spots” within the school, although this is only speculation on our part. We believe instead that fish, located in a diversity of positions within the school, get the hydrodynamic advantage of being in the group at that configuration.

We believe that fish, once they group and maintain a grouping where individuals are all within around one body length distance from each other, will necessarily get hydrodynamic benefits.As a collective group, we believe that at any one time, several different hydrodynamic mechanisms are all acting simultaneously and result in reduced energetic costs (Fig. 1).

Figure 4E: The y-axis is given in the units of 10-sec^-1 which is confusing is it 10 1/s or 1/(10s)? Why not use simply the unit of 1/s which is unambiguous?

Response: Thank you for the suggestions. We counted the turning frequency over the course of 10 seconds. To reflect more accurately on what we did, we used the suggested unit of 1/(10s) to more correctly correspond to how we made the measurements and the duration of the measurement. We recognize that this is a bit non-standard but would like to keep these units if possible.

Figure 4F: The unit in the school length is given in [mm], which suggests that the maximal measured school length is 4mm, this can't be true.

Response: Thank you for pointing this out. The unit should be [cm], which we corrected.

**Reviewer #2 (Public Review):**
Summary:This paper tests the idea that schooling can provide an energetic advantage over solitary swimming. The present study measures oxygen consumption over a wide range of speeds, to determine the differences in aerobic and anaerobic cost of swimming, providing a potentially valuable addition to the literature related to the advantages of group living.

Response: Thank you for acknowledging our contribution is a valuable addition to the literature on collective movement by animals.

Strengths:The strength of this paper is related to providing direct measurements of the energetics (oxygen consumption) of fish while swimming in a group vs solitary. The energetic advantages of schooling have been claimed to be one of the major advantages of schooling and therefore a direct energetic assessment is a useful result.

Response: Thank you for acknowledging our results are useful and provide direct measurements of energetics to prove a major advantage of schooling relative to solitary motion over a range of speeds.

Weaknesses:The manuscript suffers from a number of weaknesses which are summarised below:1. The possibility that fish in a school show lower oxygen consumption may also be due to a calming effect. While the authors show that there is no difference at low speed, one cannot rule out that calming effects play a more important role at higher speed, i.e. in a more stressful situation.

Response: Thank you for raising this creative point on “calming”. When vertebrates are moving at high speeds, their stress hormones (adrenaline, catecholamines & cortisol) increase. This phenomenon has been widely studied, and therefore, we do not believe that animals are ‘calm’ when moving at high speed and that somehow a “calming effect” explains our non-linear concave-upward energetic curves. “Calming” would have to have a rather strange non-linear effect over speed to explain our data, and act in contrast to known physiological responses involved in intense exercise (whether in fish or humans). It is certainly not true for humans that running at high speeds in a group causes a “calming effect” that explains changes in metabolic energy expenditure. We have added an explanation in the third paragraph in the section “Schooling dynamics enhances aerobic performance and reduces non-aerobic energy use”. Moreover, when animal locomotion has a high frequency of appendage movement (for both solitary individual and group movement), they are also not ‘calm’ from a behavioural point of view. Therefore, we respectfully disagree with the reviewer that the ‘calming effect’ is a major contributor to the energy saving of group movement at high speed. It is difficult to believe that giant danio swimming at 8 BL/s which is near or at their maximal sustainable locomotor limits are somehow “calm”. In addition, we demonstrated by direct energetic measurement that solitary individuals do not have a higher metabolic rate at the lower speed and thus directly show that there is very likely no cost of “uncalm” stress that would elevate the metabolic rate of solitary individuals. Furthermore, the current version of this manuscript compared the condition factor of the fish in the school and solitary individuals and found no difference (see Experimental Animal Section in the Methods). This also suggests that the measurement on the solitary fish is likely not confounded by any stress effects.

Finally, and as discussed further below, since we have simultaneous high-speed videos of fish swimming as we measure oxygen consumption at all speeds, we are able to directly measure fish behaviour. Since we observed no alteration in tail beat kinematics between schools and individuals (a key result that we elaborate on below), it’s very hard to justify that a “calming” effect explains our results. Fish in schools swimming at speed (not in still water) appear to be just as “calm” as solitary individuals.

1. The ratio of fish volume to water volume in the respirometer is much higher than that recommended by the methodological paper by Svendsen et al. (J Fish Biol 2016)Response: The ratio of respirometer volume to fish volume is an important issue that we thought about in detail before conducting these experiments. While Svendsen et al., (J. Fish Biol. 2016) recommend a respirometer volume-to-fish volume ratio of 500, we are not aware of any experimental study comparing volumes with oxygen measuring accuracy that gives this number as optimal. In addition, the Svendsen et al. paper does not consider that their recommendation might result in fish swimming near the walls of the flume (as a result of having relatively larger fish volume to flume volume) and hence able to alter their energetic expenditure by being near the wall.In our case, we needed to be able to study both a school (with higher animal volumes) and an individual (relatively lower volume) in the same exact experimental apparatus. Thus, we had to develop a system to accurately record oxygen consumption under both conditions.

The ratio of our respirometer to individual volume for schools is 693, while the value for individual fish is 2200. Previous studies (Parker 1973, Abrahams & Colgan, 1985, Burgerhout et al., 2013) that used a swimming-tunnel respirometer (i.e., a sealed treadmill) to measure the energy cost of group locomotion used values that range between 1116 and 8894 which are large and could produce low-resolution measurements of oxygen consumption. Thus, we believe that we have an excellent ratio for our experiments on both schools and solitary individuals, while maintaining a large enough value that fish don’t experience wall effects (see more discussion on this below, as we experimentally quantified the flow pattern within our respirometer).

The goal of the recommendation by Svendsen et al. is to achieve a satisfactory R2 (coefficient of determination) value for oxygen consumption data. However, Chabot et al., 2020 (DOI: 10.1111/jfb.14650) pointed out that only relying on R2 values is not always successful at excluding non-linear slopes. Much worse, only pursuing high R2 values has a risk of removing linear slopes with low R2 only because of a low signal-to-noise ratio and resulting in an overestimation of the low metabolic rate. Although we acknowledge the excellent efforts and recommendations provided by Svendsen et al., 2016, we perhaps should not treat the ratio of respirometer to organism volume of 500 as the gold standard for swim-tunnel respirometry. Svendsen et al., 2020 did not indicate how they reached the recommendation of using the ratio of respirometer to organism volume of 500. Moreover, Svendsen et al., 2020 stated that using an extended measuring period can help to resolve the low signal-to-noise ratio. Hence, the key consideration is to obtain a reliable signal-to-noise ratio which we will discuss below.

To ensure we obtain reliable data quality, we installed a water mixing loop (Steffensen et al., 1984) and used the currently best available technology of oxygen probe (see method section of Integrated Biomechanics & Bioenergetic Assessment System) to improve the signal-to-noise ratio. The water mixing loop is not commonly used in swim-tunnel respirometer. Hence, if a previously published study used a respirometer-to-organism ratio up to 8894, our updated oxygen measuring system is completely adequate to produce reliable signal-to-noise ratios in our system with a respirometer-to-organism ratio of 2200 (individuals) and 693 (schools). In fact, our original version of the manuscript used a published method (Zhang et al., 2019, J. Exp. Biol. https://doi.org/10.1242/jeb.196568) to analyze the signal-to-noise ratio and provided the quantitative approach to determine the sampling window to reliably capture the signal (Fig. S5).

1. Because the same swimming tunnel was used for schools and solitary fish, schooling fish may end up swimming closer to the wall (because of less volume per fish) than solitary fish. Distances to the wall of schooling fish are not given, and they could provide an advantage to schooling fish.

Response: This is an issue that we considered carefully in designing these experiments. After considering the volume of the respirometer and the size of the fish (see the response above), we decided to use the same respirometer to avoid any other confounding factors when using different sizes of respirometers with potentially different internal flow patterns. In particular, different sizes of Brett-type swim-tunnel respirometers differ in the turning radius of water flow, which can produce different flow patterns in the swimming section. Please note that we quantified the flow pattern within the flow tank using particle image velocimetry (PIV) (so we have quantitative velocity profiles across the working section at all tested speeds), and modified the provided baffle system to improve the flow in the working section.

Because we took high-speed videos simultaneously with the respirometry measurements, we can state unequivocally that individual fish within the school did not swim closer to the walls than solitary fish over the testing period (see below for the quantitative measurements of the boundary layer). Indeed, many previous respirometry studies do not obtain simultaneous video data and hence are unable to document fish locations when energetics is measured.

In studying schooling energetics, we believe that it is important to control as many factors as possible when making comparisons between school energetics and solitary locomotion. We took great care as indicated in the Methods section to keep all experimental parameters the same (same light conditions, same flow tank, same O2 measuring locations with the internal flow loop, etc.) so that we could detect differences if present. Changing the flow tank respirometer apparatus between individual fish and the schools studied would have introduced an unacceptable alteration of experimental conditions and would be a clear violation of the best experimental practices.

We have made every effort to be clear and transparent about the choice of experimental apparatus and explained at great length the experimental parameters and setup used, including the considerations about the wall effect in the extended Methods section and supplemental material provided.

Our manuscript provides the measurement of the boundary layer (<2.5 mm at speeds > 2 BL s-1) in the methods section of the Integrated Biomechanics & Bioenergetic Assessment System. We also state that the boundary layer is much thinner than the body width of the giant danio (~10 mm) so that the fish cannot effectively hide near the wall. Due to our PIV calibration, we are able to quantify flow near the wall.

In the manuscript, we also provide details about the wall effects and fish schools as follows from the manuscript: ”…the convex hull volume of the fish school did not change as speed increased, suggesting that the fish school was not flattening against the wall of the swim tunnel, a typical feature when fish schools are benefiting from wall effects. In nature, fish in the centre of the school effectively swim against a ‘wall’ of surrounding fish where they can benefit from hydrodynamic interactions with neighbours.”’ The notion that the lateral motion of surrounding slender bodies can be represented by a streamlined wall was also proposed by Newman et al., 1970 J. Fluid Mech. These considerations provide ample justification for the comparison of locomotor energetics by schools and solitary individuals.

1. The statistical analysis has a number of problems. The values of MO2 of each school are the result of the oxygen consumption of each fish, and therefore the test is comparing 5 individuals (i.e. an individual is the statistical unit) vs 5 schools (a school made out of 8 fish is the statistical unit). Therefore the test is comparing two different statistical units. One can see from the graphs that schooling MO2 tends to have a smaller SD than solitary data. This may well be due to the fact that schooling data are based on 5 points (five schools) and each point is the result of the MO2 of five fish, thereby reducing the variability compared to solitary fish. Other issues are related to data (for example Tail beat frequency) not being independent in schooling fish.

Response: We cannot agree with the reviewer that fish schools and solitary individuals are different statistical units. Indeed, these are the two treatments in the statistical sense: a school versus the individual. This is why we invested extra effort to replicate all our experiments on multiple schools of different individuals and compare the data to multiple different solitary individuals. This is a standard statistical approach, whether one is comparing a tissue with multiple cells to an individual cell, or multiple locations to one specific location in an ecological study. Our analysis treats the collective movement of the fish school as a functional unit, just like the solitary individual is a functional unit. At the most fundamental level of oxygen uptake measurements, our analysis results from calculating the declining dissolved oxygen as a function of time (i.e. the slope of oxygen removal). Comparisons are made between the slope of oxygen removal by fish schools and the slope of oxygen removal by solitary individuals. This is the correct statistical comparison.

The larger SD in individuals can be due to multiple biological reasons other than the technical reasons suggested here. Fundamentally, the different SD between fish schools and individuals can be the result of differences between solitary and collective movement and the different fluid dynamic interactions within the school could certainly cause differences in the amount of variation seen. Our interpretation of the ‘numerically’ smaller SD in fish schools than that of solitary individuals suggests that interesting hydrodynamic phenomena within fish schools remain to be discovered.

**Reviewer #2 (Recommendations For The Authors):**
I have reviewed a previous version of this paper. This new draft is somewhat improved but still presents a number of issues which I have outlined below.

Response: Thanks for your efforts to improve our paper with reviews, but a number of your comments apply to the previous version of the paper, and we have made a number of revisions before submitting it to eLife. We explain below how this version of the manuscript addresses many of your comments from both the previous and current reviews. As readers can see from our responses below, this version of the manuscript version no longer uses only ‘two-way ANOVA’ as we have implemented an additional statistical model. (Please see the comments below for more detailed responses related to the statistical models).

1. One of the main problems, and one of the reasons (see below) why many previous papers have measured TBF and not the oxygen consumption of a whole school, is that schooling also provides a calming effect (Nadler et al 2018) which is not easily differentiated from the hydrodynamic advantages (Abraham and Colgan 1985). This effect can reduce the MO2 while swimming and the EPOC when recovering. The present study does not fully take this potential issue into account and therefore its results are confounded by such effects. The authors state (line 401) that " the aerobic locomotion cost of solitary individuals showed no statistical difference from (in fact, being numerically lower) that of fish schools at a very low testing speed. The flow speed is similar to some areas of the aerated home aquarium for each individual fish. This suggests that the stress of solitary fish likely does not meaningfully contribute to the higher locomotor costs". While this is useful, the possibility that at higher speeds (i.e. a more stressful situation) solitary fish may experience more stress than fish in a school, cannot be ruled out.

Response: Thank you for finding our results and data useful. We have addressed the comments on calming or stress effects in our response above. The key point is that either solitary or school fish are challenged (i.e. stressed) at a high speed where the sizable increases in stress hormones are well documented in the exercise physiology literature. We honestly just do not understand how a “calming” effect could possibly explain the upward concave energetic curves that we obtained, and how “calming” could explain the difference between schools and solitary individuals. Since we have simultaneous high-speed videos of fish swimming as we measure oxygen consumption at all speeds, we are able to directly observe fish behaviour. It is not exactly clear what a “calming effect” would look like kinematically or how one would measure this experimentally, but since we observed no alteration in tail beat kinematics between schools and individuals (a key result that we elaborate on below), it’s very hard to justify that a “calming” effect explains our results. Fish in schools appear to be just as “calm” as solitary individuals.

If the reviewer's “calming effect” is a general issue, then birds flying in a V-formation should also experience a “calming effect”, but at least one study shows that birds in a V-formation experience *higher* wing beat frequencies.

In addition, Nalder et al., 2018 (https://doi.org/10.1242/bio.031997) did not study any such “calming effect”. We assume the reviewer is referring to Nalder et al., 2016, which showed that shoaling reduced fish metabolic rates in a resting respirometer that has little-to-no water current that would motivate fish to swim (which is very different from the swim-tunnel respirometer we used). Moreover, the inter-loop system used by Nalder et al., 2016 has the risk of mixing the oxygen uptake of the fish shoal and solitary individuals. Hence, we believe that it is not appropriate to extend the results of Nalder et al., 2016 to infer and insist on a calming effect for fish schools that we studied which are actively and directionally swimming over a wide speed range up to and including high speeds. Especially since our data clearly show that ‘the aerobic locomotion cost of solitary individuals showed no statistical difference from (in fact, being numerically lower) that of fish schools at very low testing speeds’. More broadly, shoaling and schooling are very different in terms of polarization as well as the physiological and behavioural mechanisms used in locomotion. Shoaling behaviour by fish in still water is not the same as active directional schooling over a speed range. Our supplementary Table 1 provides a clear definition for a variety of grouping behaviours and makes the distinction between shoaling and schooling.

Our detailed discussion about other literature mentioned by this reviewer can be seen in the comments below.

1. The authors overstate the novelty of their work. Line 29: "Direct energetic measurements demonstrating the 30 energy-saving benefits of fluid-mediated group movements remain elusive" The idea that schooling may provide a reduction in the energetic costs of swimming dates back to the 70s, with pioneering experimental work showing a reduction in tail beat frequency in schooling fish vs solitary by Zuyev, G. V. & Belyayev, V. V. (1970) and theoretical work by Weihs (1973). Work carried out in the past 20 years (Herskin and Steffensen 1998; Marras et al 2015; Bergerhout et al 2013; Hemelrijk et al 2014; Li et al 2021, Wiwchar et al 2017; Verma et al 2018; Ashraf et al 2019) based on a variety of approaches has supported the idea of a reduction in swimming costs in schooling vs solitary fish. In addition, group respirometry has actually been done in early and more recent studies testing the reduction in oxygen consumption as a result of schooling (Parker, 1973; Itazawa et al., 1978; Abrahams and Colgan 1985; Davis & Olla, 1992; Ross & Backman, 1992, Bergerhout et al 2013; Currier et al 2020). Specifically, Abrahams and Colgan (1985) and Bergerhout et al (2013) found that the oxygen consumption of fish swimming in a school was higher than when solitary, and Abrahams and Colgan (1985) made an attempt to deal with the confounding calming effect by pairing solitary fish up with a neighbor visible behind a barrier. These issues and how they were dealt with in the past (and in the present manuscript) are not addressed by the present manuscript. Currier et al (2020) found that the reduction of oxygen consumption was species-specific.

Response: We cannot agree with this reviewer that we have overstated the novelty of our work, and, in fact, we make very specific comments on the new contributions of our paper relative to the large previous literature on schooling. We are well aware of the literature cited above and many of these papers have little or nothing to do with quantifying the energetics of schooling. In addition, many of these papers rely on simple kinematic measurements which are unrelated to direct energetic measurements of energy use. To elaborate on this, we present the ‘Table R’ below which evaluates and compares each of the papers this reviewer cites above. The key message (as we wrote in the manuscript) is that none of the previous studies measured non-aerobic cost and thus do not calculate the total energy expenditure (TEE), which we show to be substantial. In addition, many of these studies do not compare schools to individuals, do not quantify both energetics and kinematics, and do not study a wide speed range. Only 33% of previous studies used direct measurements of aerobic metabolic rate to compare the locomotion costs of fish schools and solitary individuals (an experimental control). We want to highlight that most of the citations in the reviewer’s comments are not about the kinematics or hydrodynamics of fish schooling energetics, although they provide peripheral information on fish schooling in general. We also provide an overview of the literature on this topic in our paper in the Journal of Experimental Biology (Zhang & Lauder 2023 doi:10.1242/jeb.245617) and do not wish to duplicate that discussion here. We summarized and cited the relevant papers about the energetics of fish schooling in Table 1.

**Author response table 1. sa3table1:** Papers cited by Reviewer #2, and a summary of their contributions and approach.

Paper	Speed range(BL//s)	# fish inschool	TBF/phasemeasured?	Non-aerobic costmeasured?	Total energyexpendituremeasured?	Comparing fishschool & solitaryindividual
This study	0.25-8	8	YES	YES	YES	YES
Belyayev 1970	6.8	4	YES	no	no	no
Weihs 1973	Not stated	4(7)	Theoreticalmodel	no	no	no
Herskin andSteffensen1998	0.6-1.36	5	YES	no	no	no
Marras et al2015	0.83-2.5	8	YES	no	no	no
Burgerhout etal 2013	0.24-2.2	7	YES	no	no	YES
Hemelrijk et al2014	1.39	2-4	Theoreticalmodel	no	no	YES
Li et al 2021	1.2-1.6	2	YES	no	no	YES
Wiwchar et al2017	1.18-∼16	3-10	no	no	no	no
Verma et al2018	Not stated	2-3	Theoreticalmodel	no	no	no
Ashraf et al2017	0.77-3.91	5-9	YES	no	no	no
Parker, 1973	Not stated	<= 12	no	no	no	YES
Itazawa et al.,1978	No applicable,resting chamber	3 or 6	no	no	no	YES
Abrahams andColgan 1985	Not stated	6	no	no	no	no
Davis & Olla,1992	Not stated	4	no	no	no	no
Backman, 1992	Not stated	5-48	no	no	no	no
Currier et al2020	1.5-3.0	1-8	YES	no	no	YES
Halsey et al.,2018	1.1-1.4	3 or 6	YES	no	no	no
Johansen et al.,2010	0.5-2.5	4	YES	no	no	no, comparedsolitary withleading andtrailing individuals

References cited above:

Zuyev, G., & Belyayev, V. V. (1970). An experimental study of the swimming of fish in groups as exemplified by the horsemackerel [Trachurus mediterraneus ponticus Aleev]. J Ichthyol, 10, 545-549.

Weihs, D. (1973). Hydromechanics of fish schooling. Nature, 241(5387), 290-291.

Herskin, J., & Steffensen, J. F. (1998). Energy savings in sea bass swimming in a school: measurements of tail beat frequency and oxygen consumption at different swimming speeds. Journal of Fish Biology, 53(2), 366-376.

Marras, S., Killen, S. S., Lindström, J., McKenzie, D. J., Steffensen, J. F., & Domenici, P. (2015). Fish swimming in schools save energy regardless of their spatial position. Behavioral ecology and sociobiology, 69, 219-226.

Burgerhout, E., Tudorache, C., Brittijn, S. A., Palstra, A. P., Dirks, R. P., & van den Thillart, G. E. (2013). Schooling reduces energy consumption in swimming male European eels, Anguilla anguilla L. Journal of experimental marine biology and ecology, 448, 66-71.

Hemelrijk, C. K., Reid, D. A. P., Hildenbrandt, H., & Padding, J. T. (2015). The increased efficiency of fish swimming in a school. Fish and Fisheries, 16(3), 511-521.

Li, L., Nagy, M., Graving, J. M., Bak-Coleman, J., Xie, G., & Couzin, I. D. (2020). Vortex phase matching as a strategy for schooling in robots and in fish. Nature communications, 11(1), 5408.

Wiwchar, L. D., Gilbert, M. J., Kasurak, A. V., & Tierney, K. B. (2018). Schooling improves critical swimming performance in zebrafish (*Danio rerio*). Canadian Journal of Fisheries and Aquatic Sciences, 75(4), 653-661.

Verma, S., Novati, G., & Koumoutsakos, P. (2018). Efficient collective swimming by harnessing vortices through deep reinforcement learning. Proceedings of the National Academy of Sciences, 115(23), 5849-5854.

Ashraf, I., Bradshaw, H., Ha, T. T., Halloy, J., Godoy-Diana, R., & Thiria, B. (2017). Simple phalanx pattern leads to energy saving in cohesive fish schooling. Proceedings of the National Academy of Sciences, 114(36), 9599-9604.

Parker Jr, F. R. (1973). Reduced metabolic rates in fishes as a result of induced schooling. Transactions of the American Fisheries Society, 102(1), 125-131.

Itazawa, Y., & Takeda, T. (1978). Gas exchange in the carp gills in normoxic and hypoxic conditions. Respiration physiology, 35(3), 263-269.

Abrahams, M. V., & Colgan, P. W. (1985). Risk of predation, hydrodynamic efficiency and their influence on school structure. Environmental Biology of Fishes, 13, 195-202.

Davis, M. W., & Olla, B. L. (1992). The role of visual cues in the facilitation of growth in a schooling fish. Environmental biology of fishes, 34, 421-424.

Ross, R. M., Backman, T. W., & Limburg, K. E. (1992). Group-size-mediated metabolic rate reduction in American shad. Transactions of the American Fisheries Society, 121(3), 385-390.

Currier, M., Rouse, J., & Coughlin, D. J. (2021). Group swimming behaviour and energetics in bluegill Lepomis macrochirus and rainbow trout Oncorhynchus mykiss. Journal of Fish Biology, 98(4), 1105-1111.

Halsey, L. G., Wright, S., Racz, A., Metcalfe, J. D., & Killen, S. S. (2018). How does school size affect tail beat frequency in turbulent water?. Comparative Biochemistry and Physiology Part A: Molecular & Integrative Physiology, 218, 63-69.

Johansen, J. L., Vaknin, R., Steffensen, J. F., & Domenici, P. (2010). Kinematics and energetic benefits of schooling in the labriform fish, striped surfperch Embiotoca lateralis. Marine Ecology Progress Series, 420, 221-229.

1. In addition to the calming effect, measuring group oxygen consumption suffers from a number of problems as discussed in Herskin and Steffensen (1998) such as the fish volume to water volume ratio, which varies considerably when testing a school vs single individuals in the same tunnel and the problem of wall effect when using a small volume of water for accurate O2 measurements. Herskin and Steffensen (1998) circumvented these problems by measuring tailbeat frequencies of fish in a school and then calculating the MO2 of the corresponding tailbeat frequency in solitary fish in a swim tunnel. A similar approach was used by Johansen et al (2010), Marras et al (2015), Halsey et al (2018). However, It is not clear how these potential issues were dealt with here. Here, larger solitary D. aequipinnatus were used to increase the signal-to-noise ratio. However, using individuals of different sizes makes other variables not so directly comparable, including stress, energetics, and kinematics. (see comment 7 below).

Response: We acknowledge the great efforts made by previous studies to understand the energetics of fish schooling. These studies, as detailed in the table and elaborated in the response above (see comment 2) are very different from our current study. Our study achieved a direct comparison of energetics (including both aerobic and non-aerobic cost) and kinematics between solitary individuals and fish schools that has never been done before. Our detailed response to the supposed “calming effect” is given above.

As highlighted in the previous comments and opening statement, our current version has addressed the wall effect, tail beat frequency, and experimental and analytical efforts invested to directly compare the energetics between fish schools and solitary individuals. As readers can see in our comprehensive method section, achieving the direct comparison between solitary individuals and fish schools is not a trivial task.Now we want to elaborate on the role of kinematics as an indirect estimate of energetics. Our results here show that kinematic measurements of tail beat frequency are not reliable estimates of energetic cost, and the previous studies cited did not measure EPOC and those costs are substantial, especially as swimming speed increases. Fish in schools can save energy even when the tail beat frequency does not change (although school volume can change as we show). We elaborated (in great detail) on why kinematics does not always reflect on the energetics in the submitted version (see last paragraph of “Schooling dynamics and energy conservation” section). Somehow modeling what energy expenditure should be based only on tail kinematics is, in our view, a highly unreliable approach that has never been validated (e.g., fish use more than just tails for locomotion). Indeed, we believe that this is an inadequate substitute for direct energy measurements. We disagree that using slightly differently sized individuals is an issue since we recorded fish kinematics across all experiments and included the measurements of behaviour in our manuscript. Slightly altering the size of individual fish was done on purpose to provide a better ratio of respirometer volume to fish volume in the tests on individual fish, thus we regard this as a benefit of our approach and not a concern.

Finally, in another study of the collective behaviour of flying birds (Usherwood, J. R., Stavrou, M., Lowe, J. C., Roskilly, K. and Wilson, A. M. (2011). Flying in a flock comes at a cost in pigeons. Nature 474, 494-497), the authors observed that wing beat frequency can increase during flight with other birds. Hence, again, we cannot regard movement frequency of appendages as an adequate substitute for direct energetic measurements.

1. Svendsen et al (2016) provide guidelines for the ratio of fish volume to water volume in the respirometer. The ratio used here (2200) is much higher than that recommended. RFR values higher than 500 should be avoided in swim tunnel respirometry, according to Svendsen et al (2016).

Response: Thank you for raising this point. Please see the detailed responses above to the same comment above. We believe that our experimental setup and ratios are very much in line with those recommended, and represent a significant improvement on previous studies which use large ratios.

1. Lines 421-436: The same goes for wall effects. Presumably, using the same size swim tunnel, schooling fish were swimming much closer to the walls than solitary fish but this is not specifically quantified here in this paper. Lines 421-436 provide some information on the boundary layer (though wall effects are not just related by the boundary layer) and some qualitative assessment of school volume. However, no measurement of the distance between the fish and the wall is given.

Response: Please see the detailed responses above to the same comment. Specifically, we used the particle image velocimetry (PIV) system to measure the boundary layer (<2.5 mm at speeds > 2 BL s-1) and stated the parameters in the methods section of the Integrated Biomechanics & Bioenergetic Assessment System. We also state that the boundary layer is much thinner than the body width of the giant danio (~10 mm) so that the fish cannot effectively hide near the wall. Due to our PIV calibration, we are able to quantify flow near the wall.

Due to our video data obtained simultaneously with energetic measurements, we do not agree that fish were swimming closer to the wall in schools and also note that we took care to modify the typical respirometer to both ensure that flow across the cross-section did not provide any refuges and to quantify flow velocities in the chamber using particle image velocimetry. We do not believe that any previous experiments on schooling behaviour in fish have taken the same precautions.

1. The statistical tests used have a number of problems. Two-way ANOVA was based on school vs solitary and swimming speed. However, there are repeated measures at each speed and this needs to be dealt with. The degrees of freedom of one-way ANOVA and T-tests are not provided. These tests took into account five groups of fish vs. five solitary fish. The values of MO2 of each school are the result of the oxygen consumption of each fish, and therefore the test is comparing 5 individuals (i.e. an individual is the statistical unit) vs 5 schools (a school made out of 8 fish is the statistical unit). Therefore the test is comparing two different statistical units. One can see from the graphs that schooling MO2 tend to have a smaller SD than solitary data. This may well be due to the fact that schooling data are based on 5 points (five schools) and each point is the result of the MO2 of five fish, thereby reducing the variability compared to solitary fish. TBF, on the other hand, can be assigned to each fish even in a school, and therefore TBF of each fish could be compared by using a nested approach of schooling fish (nested within each school) vs solitary fish, but this is not the statistical procedure used in the present manuscript. The comparison between TBFs presumably is comparing 5 individuals vs all the fish in the schools (6x5=30 fish). However, the fish in the school are not independent measures.

Response: We cannot agree with this criticism, which may be based on this reviewer having seen a previous version of the manuscript. We did not use two-way ANOVA in this version. This version of the manuscript reported the statistical value based on a General Linear Model (see statistical section of the method). We are concerned that this reviewer did not in fact read either the Methods section or the Results section. In addition, it is hard to accept that, from examination of the data shown in Figure 3, there is not a clear and large difference between schooling and solitary locomotion, regardless of the statistical test used.

Meanwhile, the comments about the ‘repeated’ measures from one speed to the next are interesting, but we cannot agree. The ‘repeated’ measures are proper when one testing subject is assessed before and after treatment. Going from one speed to the next is not a treatment. Instead, the speed is a dependent and continuous variable. In our experimental design, the treatment is fish school, and the control is a solitary individual. Second, we never compared any of our dependent variables across different speeds within a school or within an individual. Instead, we compared schools and individuals at each speed. In this comparison, there are no ‘repeated’ measures.We agree with the reviewer that fish in the school are interacting (not independent). This is one more reason to support our approach of treating fish schools as a functional and statistical unit in our experiment design (more detailed responses are stated in the response to the comment above).

1. The size of solitary and schooling individuals appears to be quite different (solitary fish range 74-88 cm, schooling fish range 47-65 cm). While scaling laws can correct for this in the MO2, was this corrected for TBF and for speed in BL/s? Using BL/s for speed does not completely compensate for the differences in size.

Response: Our current version has provided justifications for not conducting scaling in the values of tail beat frequency. Our justification is “The mass scaling for tail beat frequency was not conducted because of the lack of data for D. aequipinnatus and its related species. Using the scaling exponent of distant species for mass scaling of tail beat frequency will introduce errors of unknown magnitude.”. Our current version also acknowledges the consideration about scaling as follows: “Fish of different size swimming at 1 BL s-1 will necessarily move at different Reynolds numbers, and hence the scaling of body size to swimming speed needs to be considered in future analyses of other species that differ in size”

**Reviewer #3 (Public Review):**
Summary:Zhang and Lauder characterized both aerobic and anaerobic metabolic energy contributions in schools and solitary fishes in the Giant danio (Devario aequipinnatus) over a wide range of water velocities. By using a highly sophisticated respirometer system, the authors measure the aerobic metabolisms by oxygen uptake rate and the non-aerobic oxygen cost as excess post-exercise oxygen consumption (EPOC). With these data, the authors model the bioenergetic cost of schools and solitary fishes. The authors found that fish schools have a J-shaped metabolism-speed curve, with reduced total energy expenditure per tail beat compared to solitary fish. Fish in schools also recovered from exercise faster than solitary fish. Finally, the authors conclude that these energetic savings may underlie the prevalence of coordinated group locomotion in fish.The conclusions of this paper are mostly well supported by data, but some aspects of methods and data acquisition need to be clarified and extended.

Response: Thank you for seeing the value of our study. We provided clarification of the data acquisition system with a new panel of pictures included in the supplemental material to show our experimental system. We understand that our methods have more details and justifications than the typical method sections. First, the details are to promote the reproducibility of the experiments. The justifications are the responses to reviewer 2, who reviewed our previous manuscript version and also posted the same critiques after we provided the justifications for the construction of the system and the data acquisition.

Strengths:This work aims to understand whether animals moving through fluids (water in this case) exhibit highly coordinated group movement to reduce the cost of locomotion. By calculating the aerobic and anaerobic metabolic rates of school and solitary fishes, the authors provide direct energetic measurements that demonstrate the energy-saving benefits of coordinated group locomotion in fishes. The results of this paper show that fish schools save anaerobic energy and reduce the recovery time after peak swimming performance, suggesting that fishes can apport more energy to other fitness-related activities whether they move collectively through water.

Response: Thank you. We are excited to share our discoveries with the world.

Weaknesses:Although the paper does have strengths in principle, the weakness of the paper is the method section. There is too much irrelevant information in the methods that sometimes is hard to follow for a researcher unfamiliar with the research topic. In addition, it was hard to imagine the experimental (respirometer) system used by the authors in the experiments; therefore, it would be beneficial for the article to include a diagram/scheme of that respiratory system.

Response: We agree with the reviewer and hence added the pictures of the experimental system in the supplementary materials (Fig. S4). We think pictures are more realistic to present the system than schematics. We also provide a picture of the system during the process of making the energetic measurements. It is to show the care went to ensure fish are not affected by any external stimulation other than the water velocity. The careful experimental protocol is very critical to reveal the concave upward shaped curve of bony fish schools that was never reported before. Many details in the methods have been included in response to Reviewer 2.

**Reviewer #3 (Recommendations For The Authors):**
Overall, this is a very interesting, well-written, and nice article. However, many times the method section looks like a discussion. Furthermore, the authors need to check the use of the word "which" throughout the text. I got the feeling that it is overused/misused sometimes.

Response: Thank you for the positive comments. The method is written in that way to address the concerns of Reviewer 2 who reviewed our previous versions. We corrected the overuse of ‘which’ throughout the manuscript.